# Using Deep Reinforcement Learning with Automatic Curriculum Learning for Mapless Navigation in Intralogistics

**Honghu Xue** [1,*], **Benedikt Hein** [1,2,†], **Mohamed Bakr** [2], **Georg Schildbach** [3], **Bengt Abel** [2] and **Elmar Rueckert** [4]

1    Institute for Robotics and Cognitive Systems, University of Luebeck, 23562 Luebeck, Germany;
     benedikt.hein@hsu-hh.de
2    KION Group AG, Technology and Innovation, 22113 Hamburg, Germany; mohamed.bakr@still.de (M.B.);
     bengt.abel@still.de (B.A.)
3    Institute for Electrical Engineering in Medicine, University of Luebeck, 23562 Luebeck, Germany;
     georg.schildbach@uni-luebeck.de
4    Institute for Cyber Physical Systems, Montanuniversität Leoben, 8700 Leoben, Austria;
     rueckert@unileoben.ac.at
*    Correspondence: xue@rob.uni-luebeck.de
†    Current address: Institute of Automation Technology, Helmut Schmidt University, 22043 Hamburg, Germany.

**Abstract:** We propose a deep reinforcement learning approach for solving a mapless navigation problem in warehouse scenarios. In our approach, an automatic guided vehicle is equipped with two LiDAR sensors and one frontal RGB camera and learns to perform a targeted navigation task. The challenges reside in the sparseness of positive samples for learning, multi-modal sensor perception with partial observability, the demand for accurate steering maneuvers together with long training cycles. To address these points, we propose *NavACL-Q* as an automatic curriculum learning method in combination with a distributed version of the soft actor-critic algorithm. The performance of the learning algorithm is evaluated exhaustively in a different warehouse environment to validate both robustness and generalizability of the learned policy. Results in NVIDIA Isaac Sim demonstrates that our trained agent significantly outperforms the map-based navigation pipeline provided by NVIDIA Isaac Sim with an increased agent-goal distance of 3 m and a wider initial relative agent-goal rotation of approximately 45°. The ablation studies also suggest that *NavACL-Q* greatly facilitates the whole learning process with a performance gain of roughly 40% compared to training with random starts and a pre-trained feature extractor manifestly boosts the performance by approximately 60%.

**Keywords:** deep reinforcement learning; automatic curriculum learning; autonomous navigation; multi-modal sensor perception

## 1. Introduction

Mobile robot navigation has received broad applications and has been intensively studied in recent decades, ranging from urban driving [1,2] to indoor navigation [3]. One popular approach is Simultaneous Localization and Mapping (SLAM) [4] via a combination of various algorithms. In the SLAM procedure, the map is generated via sensors, and planning algorithms [5] are used on top of the map. Nonetheless, the limitations are also manifest. In particular, the efforts to build a map can be expensive in case of dynamic environments. Usually, disparate sensory sources are necessary for non-stationary environment, which additionally requires sensor fusion [6,7], complicating the process. The generated map accuracy also plays a vital role for navigation quality and to generate a sufficiently accurate map, extra human engagements for data acquisition are entailed [8].

On the other hand, Deep Reinforcement Learning (DRL) has found successful application in games [9,10] and robotic applications such as robot manipulation [11,12] and navigation [13,14] by combining the power of Deep Neural Network (DNN) and Reinforcement Learning (RL) [15]. The component RL serves as an approach for optimal decision

making based on a Markov decision process, where the agent learns to act given the observations in a loop to maximize the long-term utility. DNNs empower the RL with extension to high-dimensional observations, for instance, visual data, LiDAR readings and etc. One major appealing point of DRL is the ability to learn from scratch without expert demonstration, which makes DRL an end-to-end learning approach. A second benefit lies in its non-reliance on a transition model (model-free). The agent learns via interactions with the environment in a trial-and-error manner. In contrast, optimal control algorithms like model predictive control [16] necessitate a carefully derived physical model, which can be demanding for computing. This could be particularly helpful in the case of multi-sensor observations, where it is extremely sophisticating to manually define the rules for sensor fusion and to calculate transition dynamics. We give an illustration of previous works on model-free DRL algorithms in Section 2.

In our work, we aim to address a navigation problem in a warehouse scenario, where the Automatic Guided Vehicle (AGV) aims to navigate underneath a dolly purely relying on its own sensor readings. The mobile robot is equipped with frontal Red-Green-Blue (RGB) camera and two LiDAR sensors measuring the distance, illustrated in Figure 1. Based on these two types of sensor readouts, i.e., multi-modal sensor perceptions, the AGV is supposed to steer towards the target. In a typical warehouse setting, the environment is non-stationary, where the location of the target and obstacles are subject to change. In this work, we are especially interested in the ability of DRL to directly map the agent's multi-modal sensory reading to the control commands via neural networks, without the efforts to generate a map or human demonstrations or manually processing multi-modal sensor fusions. Moreover, it is desired that the learned strategy shows generalizability with respect to different interior settings, e.g., room sizes, position of the obstacles, etc.

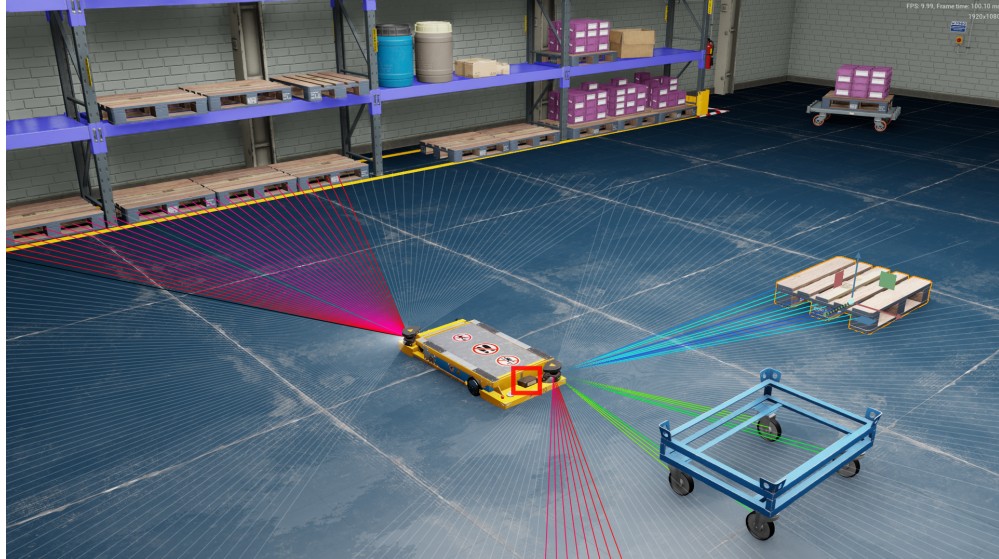

**Figure 1.** Illustration of the dolly (blue) and the robot in our simulated warehouse environment. The lines connected to the robot's chassis visualize the LiDAR distance measuring beams. In this figure, NVIDIA Omniverse™ [17] is used for visualization. The front-facing camera is placed right in the center of the chassis of the vehicle, highlighted by the red square and captures images with a resolution of $80 \times 80$ pixels. Two additional LiDAR sensors are placed at the diagonal corners of the vehicle, with each emitting 128 beams and covering a field of view of $225°$, respectively.

Formulating the navigation task fulfilling the aforementioned criteria as a DRL problem introduces a lot of difficulties. A first challenge is the sparseness of positive samples, where the sparseness stems from the low likelihood of reaching a constricted goal space (underneath the dolly). It is shown in [18] that DRL algorithms learn a robust policy only when both sufficient positive and negative samples are provided for learning. A second challenge is the multi-modal sensor perception together with partial observability, where

the mobile agent may not perceive the target given only the frontal RGB camera and therefore loses the goal information. The robot needs to learn to behave rationally to search for the goal and to infer whether the goal is present merely from its own sensory readings. Moreover, DRL algorithms converge to a reasonable performance after huge amounts of interaction experience [19], resulting in a long training cycle. Hence, we investigate potential approaches to reduce the overall training duration without compromising the reward design that can facilitate the training, which is only feasible in simulation but not in real application. It is noteworthy that parking under the dolly is demanding as it requires accurate steering maneuvers and the robot directly learns the low-level differential drive command instead of a set of pre-defined movement primitives.

To address these challenges, we proposed a distributed version of Soft Actor-Critic with automatic curriculum learning (ACL) to increase the number of positive samples the and to reduce the overall training cycle. We extend one ACL algorithm *NavACL* [20] to a more general case, named as *NavACL-Q*. The performance of the learned policy is also systematically evaluated in a different testing scenario for robustness and generalizability check. The ablation studies are conducted to check the effects of a pre-trained feature extractor and ACL on the performance gain, respectively. We finally show that our approach outperforms a baseline map-based navigation pipeline provided by Nvidia SDK [21].

## 2. Related Work

In this section, we present an overview on the recent progress of DRL algorithms in Section 2.1 and their applications in navigation tasks in Section 2.2. Moreover, previous work on curriculum learning on RL tasks are also investigated and studied in Section 2.3.

### 2.1. Model-Free Deep Reinforcement Learning Algorithms

Model-free DRL algorithms have become increasingly successful in solving complex tasks featuring high dimensional observations without the need of knowing the transition dynamics of the environment. In the first impressive work, Deep Q-network (DQN) [19], an agent was trained to play Atari video games and reached human-level performance. They applied a DNN to map raw-pixel visual input to the corresponding Q-values. The work introduced a frozen target network to alleviate the deadly-triad problem [15,22]. Another major contribution is the usage of an experience replay buffer to decorrelate the temporal dependence between samples within one episode, therefore enhancing the performance. These components are widely used in other off-policy DRL algorithms.

There are several improvements proposed to enhance the performance of DQNs. Double deep Q-networks [23] address the problem of the maximization bias analogously to Double Q-learning [15]. Noisy networks [24] improves the exploration strategy of the agent by replacing the standard $\epsilon$-greedy algorithms by the noisy networks, where the weights of network are injected with zero-mean Gaussian noises, resulting in randomness in choosing the action.

All previous methods are designed for discrete action spaces. Other approaches generalize to continuous action space. These algorithms, so-called Policy-Gradient (PG) methods, have an additional learnable component, *actor*, which maps states to actions maximizing the return. The on-policy algorithm asynchronous advantage actor-critic [25] reduces the variance on actor learning compared to REINFORCE [15] and reduces the overall training cycle by having multiple threads collecting the experience in parallel. Asynchronous advantage actor-critic outperforms vanilla DQN on Atari Games. Proximal policy optimization [26] tries to achieve monotonic policy improvements while avoiding a large change of the policy that could cause performance collapse. It updates the policy by additionally penalizing the KL-divergence between previous policy and the new policy.

Despite the success of on-policy PG algorithms, they are not as sample-efficient as off-policy variants [27]. This disadvantage becomes more apparent in case of an expensive simulator. The off-policy policy algorithm deep deterministic policy gradient [28] extends

DQN to the continuous action case. The algorithm Twin-Delayed deep deterministic policy gradient [29] further improves deep deterministic policy gradient by addressing maximization bias and proposes to add noise to the action with delayed policy update for a more stable training.

However, the shortage of deep deterministic policy gradient and twin-delayed deep deterministic policy gradient is that the exploration scheme must be done explicitly and that they can only model deterministic optimal policies. In their original work, they applied Gaussian noise to enable exploration. A sufficient exploration is crucial for the final performance for any RL algorithm. However, in contrast to some explicit exploration strategies [10,30–32], the work in [33,34] proposed a new category of RL algorithms, maximal-entropy reinforcement learning, in particular, the Soft Actor-Critic (SAC) algorithm. SAC tries to address the exploration problem by incorporating the entropy of policy as an exploration bonus into the return, equivalent to an implicit exploration schedule. A second benefit of SAC is the ability to model multi-modal optimal policies with a probabilistic characterization. SAC was reported to outperform (twin-delayed) deep deterministic policy gradient in some continuous control problems in Mujoco [35], e.g., Half Cheetah, Humanoid.

In our task of intralogistics navigation, the mobile robot requires accurate steering abilities, i.e., continuous action commands, to navigate beneath the target dolly. Moreover, the agent only shows signs of learning with the presence of adequate successful trials, which requires sufficient exploration in the environment. For these reasons, we choose SAC for our use case.

### 2.2. Deep Reinforcement Learning for Robot Navigation Tasks

There has been extensive research in robot indoor navigation tasks [36–38] using SLAM. The work [39] focuses on improving the navigation accuracy of AVG in an indoor environment via SLAM, i.e., first positioning the AVG via multiple sensor readouts and then performing navigation. They improve the positioning accuracy via multiple sets of measurements and multilaterations. DRL has been investigated for the task of robot navigation in recent years. The contribution of [40] proposes virtual-to-real DRL for mapless navigation on mobile robots with continuous actions. In their work, the mobile robots acquire LiDAR observations, relative angles and distances from the current robot pose to the goal. They trained the agent using asynchronous advantage actor-critic and demonstrated both success and generalizability in new environments in Gazebo simulator [41]. However, their state and reward formulation is impractical for training directly in real environments, as they assume the knowledge of goal position and current robot pose, which are expensive to acquire in the real world. The work of [42] explores the potential of using discrete actions for navigating and they adopted the similar problem setting as [40]. In their findings, training discrete action space using double deep Q-network and PER is more efficient than a continuous action space via deep deterministic policy gradient and proximal policy optimization. However, their approach has restricted the degree of freedom in trajectory space due to the choice of discrete actions and also encounters the same problem in real application as [40]. The authors of [43] apply the similar problem formulation on a multi-agent scenario, where a swarm of robots learn to navigate to their own targets (in group formation) without colliding with each other. Despite the impressive result in simulation, the information on relative pose to goal is still required. We also see such settings in [44,45]. Some other work [46] also proposes deep learning approaches for traffic flow prediction.

Some other work implements a target-driven approach for visual navigation, where an image of the target is also provided as a part of the observation. In [47], they used a pre-trained network ResNet-50 [48] to transform both the current and target visual observation into the embedding space and afterwards mapped to policy and critic values. Their work mainly addresses the generalizability among different scenes and the learned agent demonstrates the ability to reach manifold targets in various interior environments. However, the actor outputs only four discrete high-level actions, which greatly alleviates

the difficulty of DRL training. The work of [49] also exploits the same idea with two major improvements. Firstly, they resorted to additional auxiliary tasks, e.g., learning meaningful segmentation and reward prediction, for performance boost, where the convolutional encoders are learned end-to-end. Secondly, they mitigated partial observability by keeping a longer historical observation using long short-term memory (LSTM) [50] instead of frame stacking. The performance boost could be seen with their proposed approach. Both of the two works used asynchronous advantage actor-critic as DRL algorithm.

A third category implements map-based DRL, where the map is either given or generated online. The work of [51] generates egocentric local occupancy maps for local collision avoidance via SLAM. A second component local planner proposes local goals given the final target position. This is ensued by a DRL algorithm that maps the agent's velocity, the planned local goal and local occupancy maps to 28 discrete actions. They used dueling double deep Q-network [52] with PER and randomized the number of obstacles and initial position to facilitate learning. However, their problem formulation only enables robot to navigate to the local goal instead of the final target, which greatly alleviates the difficulty in RL, but heavily relies on the quality of SLAM and the local planner. In comparison, our approach does not require any complicated SLAM-related information or any local planners. It only resorts to multi-modal sensor readouts, fuses them, and maps to continuous control commands for reaching the final goal in a blackbox fashion, where we purely rely on the power of DNNs.

### 2.3. Curriculum Learning for Reinforcement Learning Tasks

One main challenge of RL is that it requires prohibitive number of interactions steps with the environment to reach a reasonable convergence. Moreover, it is also crucial that the agent keeps a reasonable proportion of the positive experience leading to high returns and negative experiences with low returns so as to grant the agent a effective learning signal. In our navigation task, where the robot has to go through a long time-horizon to reach its target state, the probability of positive experiences, i.e., reaching goal state, is merely marginal. In such settings, the agent suffers severely from the class imbalance problem and will mostly learn from negative experience, only avoiding obstacles but failing to arrive at the goal. One solution is to resort to expert demonstrations. Nevertheless, it breaks the nice property of learning from scratch. In some challenging tasks, it is even hard for a human to demonstrate. In this work, we focus on learning from scratch. The second alternative is Curriculum Learning (CL). It proposes a set of curricula (intermediate tasks) starting from easy tasks and progressively increasing the task difficulty until the desired task is solved. With such curricula, the agent is more likely to get positive experience from easy tasks and can transfer the gained knowledge to the upcoming tasks, which decreases the overall training time as compared to directly learning from scratch on a hard task [53]. For these reasons, we also apply CL together with DRL for our case.

The term Curriculum Learning was first proposed by [54]. They find providing an ordered sequence of the training samples rather than random sequence can facilitate the learning speed and generalizability. Such ideas were present in Prioritized Experience Replay (PER) [55], where the samples with high TD-errors get higher priorities to be sampled. This is equivalent to an implicit curriculum on the samples. PER is reported to have better performance than normal replay buffer in DQN. Alternatives for different definition of priorities on sample-level are also presented in [56], where they further considered a user-defined self-paced priority function and a coverage function to avoid repetitive sampling of only high priority samples. In [57], the PER is extended in another manner by using a network to predict the significance of each training sample. It can therefore even predict the importance of unseen training samples.

The above work mainly proposes various heuristics to reach sample-level curriculum learning. Other work involves how to generate intermediate tasks, how the tasks can be sequence properly to accelerate training and how to transfer the knowledge between tasks. In [58], a number of methods are introduced to create intermediate tasks with the

assumption that all the tasks can be parameterized as a vector of features. The overall process is incrementally developing subtasks that are dependent on the trajectories of learned policies and the current tasks. They propose several heuristics to generate new subtasks, i.e., *Task Dimension Simplification*, *Promising Initializations* to deal with sparse-reward signals, *Mistake-Driven Subtasks* with a focus on avoiding unwanted behaviors etc. Hindsight Experience Replay (HER) [59] forms a curriculum by storing additional trajectories with imaginary goal states as training samples. HER incorporates the goal state $g$ together with the current state $s$ to learn the value function $v_\pi(s, g)$. The target task may be originally hard to achieve, but positive experience could be easily obtained when the goal state is changed to the terminal state of this episode. Relying on the expressiveness of DNNs, the policy learned from the ever-changing goal states can be beneficial for generalizing to the desired goal tasks. The algorithm Curriculum-guided HER (CHER) [60] improves the PER by adaptively selecting the imaginary goal state. The selection criteria are goal diversity and proximity to the desired goal state. The curriculum is created by initially allowing for diverse imaginary goal sets and gradually converging to the proximity to the true goal. However, these HER-variants necessitate the explicit knowledge of the goal state and the fictitious reward function for arbitrary goal states.

Some other work defines the curriculum by generating a set of initial states instead of goal states. The work of [61] proposes reverse curriculum generation, where the distribution of the initial states become farther away from the goal states. Candidates of initial states for the next episode are generated by random walk from the existing starting states. To select the exact starting state, the expected return for these candidates is computed and the one lying in the pre-defined interval is selected. The approach in [62] shares a similar idea, whereas it generates the candidate starting states not by random walk but via approximated transition dynamics, i.e., estimating the number of steps to reach the goal. They sampled from a mixture of successfully-trained tasks and new candidates to avoid catastrophic forgetting.

Our work also generalizes a recent ACL approach *NavACL* [20]. *NavACL* generates a set of curricula based on a parallelly-learned success prediction network that estimates the probability of the agent to reach goal given the current policy. In the original work, *NavACL* is reported to greatly improve the whole learning procedure in terms of success probability of the navigation task. The details are presented in Section 3.4.

## 3. Materials and Methods

The task of load carrier docking in the context of intralogistics considers the targeted navigation of a transport robot underneath a target dolly. A first challenge of our task is to learn accurate steering commands so as to reach a very constricted goal space, where the area underneath the target dolly is deemed as the goal. We provide the detailed specification of navigation vehicle and the target dolly together with the simulation environment in Section 3.2 for a direct view on how challenging the task is. In a goal-reaching task, one key for any DRL algorithm to reach a reasonable performance is that the agent has an adequate number of successful trails. A first solution is to deploy efficient exploration strategy, where we used soft actor-critic [34], introduced in Section 3.1. With a more informative exploration strategy, the agent will reach the target given sufficient number of trials. Soft actor-critic also features continuous action output and for accurate control. However, the agent behaves more exploratorily at the onset of training. The majority of explorative trials can end up with failure, interpreted as negative experience, especially when the required time horizon for reaching the goal is long and a constricted goal space. This results in sparseness of positive experience, potentially making DRL fail in learning. Therefore, we apply automatic curriculum learning to increase the probability of positive experience by starting training from easy tasks. We elaborate our proposed automatic curriculum learning algorithm *NavACl-Q* as a generalization of *NavACl* in Section 3.4. To further accelerate DRL training, we first parallelize multiple agents collecting the experience, giving rise to a distributed soft actor-critic. Moreover, some ablation variants are conducted to examine if the performance

can be further enhanced, shown in Section 3.5. A complete formulation of the intralogistics navigation task as a DRL problem and algorithm hyperparameters are also elaborated in Section 3.3.

### 3.1. Maximum Entropy Reinforcement Learning–Soft Actor-Critic

An RL problem can be seen as a sequential decision problem in a Markov Decision Process (MDP). It is defined as a tuple $(S, A, P, R, \gamma)$, where $S$ and $A$ are, respectively, the set of states and actions, $P$ denotes state transition probability matrix, specifically, the probability of transiting to a successor state $s_{t+1}$ from the state $s_t$ by taking action $a_t$. The reward function $R : S \times A \to \mathbb{R}$, which returns a number indicating the utility of performing an action in the current state. The last element is the discount factor $\gamma \in [0, 1]$, which leverages the importance on short-term rewards against long-term rewards.

In RL, the agent interacts with the environment to collect experience and tries to learn an optimal policy $\pi^*$ such that the return, namely cumulative reward, is maximized.

$$\pi^* = \arg\max_{\pi} \mathbb{E}_{\tau \sim \pi} \left[ \sum_{t=0}^{T} \gamma^t r_{t+1} \right],$$

where $r_t$ refers to the *immediate reward* at time point $t$, $\tau$ is the trajectory, characterized as a sequence of $\{s_0, a_0, r_1, \ldots, s_T, a_T, r_{T+1}\}$ following the policy $\pi$ and $T$ is the time horizon to reach the terminal state.

The maximum entropy RL algorithm Soft Actor Critic (SAC) [33,34] differs from the standard RL in that it changes the goal by incorporating an additional weighted policy entropy term $\mathcal{H}(\pi(\cdot \mid s_t))$, The policy entropy describes the stochasticity of the policy, indicating the degree of exploration. In this manner, the greedy policy returned by SAC includes an internal exploration bonus, which is advantageous for maximum entropy RL, as no explicit exploration strategy needs to be formulated. The objective function of SAC is defined as:

$$\pi^* = \arg\max_{\pi} \mathbb{E}_{\tau \sim \pi} \left[ \sum_{t=0}^{T} \gamma^t (r_{t+1} + \alpha \mathcal{H}(\pi(\cdot \mid s_t))) \right],$$

where $\alpha$ is the temperature coefficient determining the weight for policy entropy. In [33], the weights are pre-determined by the users and need manual tunning for different tasks. As an improvement, the work [34] proposed automatically adjusting $\alpha$ for different tasks. To enable a learnable $\alpha$, the authors impose a constraint that $\mathbb{E}_{(\mathbf{s}_t, \mathbf{a}_t) \sim \rho_\pi}[\mathcal{H}(\pi(\cdot \mid s_t))] \geq \overline{\mathcal{H}}$, where $\overline{\mathcal{H}}$ is a pre-defined entropy lower bound to ensure a minimal level of exploration. The constraint can be cast into a dual optimization problem shown below so that the temperature $\alpha$ turns a learnable parameter. For an exact theoretical derivation, please refer to the original work [34].

$$\alpha_t^* = \arg\min_{\alpha_t} \mathbb{E}_{\mathbf{a}_t \sim \pi_t^*} \left[ -\alpha_t \log \pi_t^*(\mathbf{a}_t \mid \mathbf{s}_t; \alpha_t) - \alpha_t \overline{\mathcal{H}} \right].$$

The critic part learns the Q-values with the additional policy entropy term based on a re-defined Bellman update:

$$\hat{Q}(\mathbf{s}_t, \mathbf{a}_t) = r_{t+1} + \gamma \left( Q_{\tilde{\phi}}(s_{t+1}, \tilde{a}_{t+1}) - \alpha \log \pi_\theta(\tilde{a}_{t+1} \mid s_{t+1}) \right),$$

where $\tilde{a}_{t+1} \sim \pi_\theta(\cdot \mid s_{t+1})$, $\phi$ and $\tilde{\phi}$ refer, respectively, to the running and target critic network for training stability similar to DQN. The critic loss is then computed by sampling a minibatch from the replay buffer $\mathcal{D}$.

$$J_Q(\theta) = \mathbb{E}_{(\mathbf{s}_t, \mathbf{a}_t) \sim \mathcal{D}} \left[ \frac{1}{2} \left( Q_\phi(\mathbf{s}_t, \mathbf{a}_t) - \hat{Q}(\mathbf{s}_t, \mathbf{a}_t) \right)^2 \right].$$

In the policy improvement step, the actor is updated towards an exponential of the Q-values to still allow for a distribution of the policy via Kullback–Leibler divergence.

$$J_\pi(\phi) = \mathbb{E}_{s_e \sim \mathcal{D}}\left[ D_{\mathrm{KL}}\left( \pi_\theta(\cdot \mid \mathbf{s}_t) \middle\| \frac{\exp\left(Q_\phi(\mathbf{s}_t, \cdot)\right)}{Z_\theta(\mathbf{s}_t)} \right) \right].$$

In our implementation, we also apply similar tricks as Double Q-learning [63,64] to avoid maximization bias. Furthermore, SAC with a learnable temperature coefficient $\alpha$ [34] is used, as a fixed one requires good domain knowledge which is assumed to be unknown in most cases.

### 3.2. Simulation Environment

We run our experiments on the simulator NVIDIA Isaac SDK$^{\mathrm{TM}}$ [21]. The target dolly consists of a steel frame that can be loaded with a pallet. The dolly stands on four passive wheels, which makes it transportable. Figure 1 illustrates the mobile robot and the dolly used for this paper. The simulated vehicle is a platform robot which is specifically built for load carrier docking and is actuated by a differential drive. The vehicle and dolly specification is shown in Appendix D. It is noteworthy that the width of the dolly is only 21 cm wider than vehicle so that very accurate steering efforts are required to successfully navigate underneath the dolly, corresponding to a constricted goal space.

### 3.3. Reinforcement Learning Problem Setup

Here we present how the AVG navigation problem can be formulated as an DRL problem. In this work, the observation space $O$ is defined as a concatenation of $[O_v, O_l, O_{ar}]$. To deal with partial observability, we stack 4 most recent RGB image $O_v$ along the channel dimension exactly as how DQN processed Atari games [19]. The second observation component is LiDAR observation $O_l$, since it mostly reaches fully-observability, we just retain the most recent LiDAR readings. To further increase the information content, we also keep the same length of historical actions and rewards as a part of observation similarly to [65]. Note that no additional handcrafted high-level information, e.g., the position of the robot or the dolly is given. Furthermore, neither a method for localization nor mapping is used. The complete state design is summarized in Table 1. The visual perception $O_v$ and the LiDAR readings $O_l$ are rescaled to $[0, 1]$. All these post processed features serve as input to critic and actor network in SAC, as shown in Figure 2.

**Table 1.** Summary of the sensory observations and additional statistics that describe the state design of this thesis.

| Observation Components | |
|---|---|
| **Description** | **Dimensions** |
| Sequence of the four most recent camera RGB images | $\mathbf{R}^{4 \times 3 \times 80 \times 80}$ |
| Current LiDAR sensor input (front and back sensor concatenated) | $\mathbf{R}^{1 \times 256}$ |
| History of the four previously taken actions | $\mathbf{R}^{4 \times 2}$ |
| History of the four previously received rewards | $\mathbf{R}^{4 \times 1}$ |

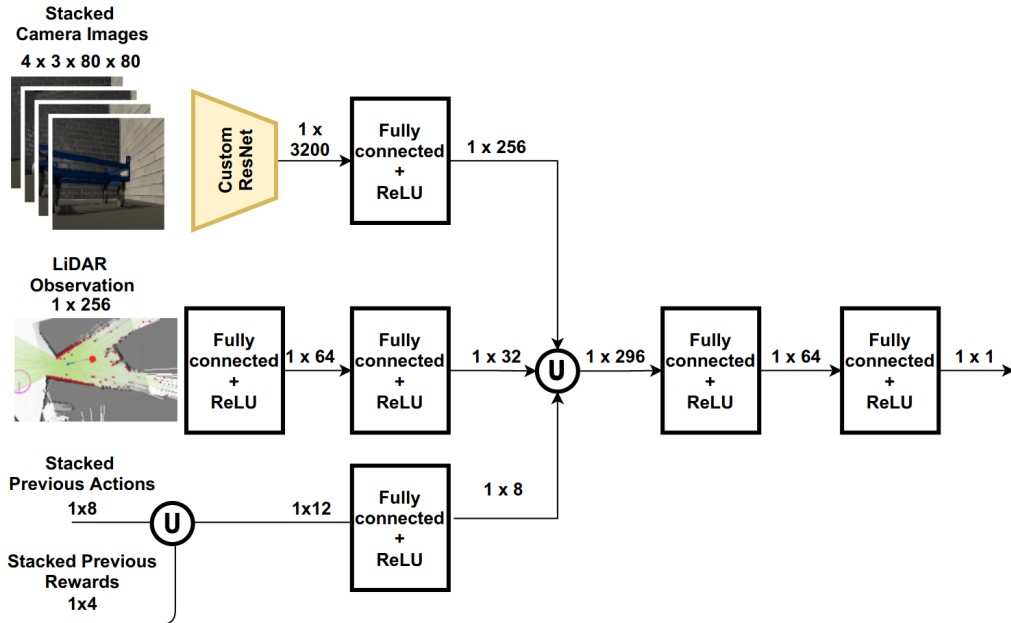

**Figure 2.** An illustration of the Critic network architecture, consisting of ResNet blocks [48] for feature extraction (highlighted by the yellow shape) and fully-connected layers for LiDAR inputs and historical action and rewards. We concatenate the outputs of the three parts (illustrated by the ∪ symbol) to establish a learned sensor fusion. For actor part, only the output layer is changed. The details of ResNet blocks are shown in Appendix A.

The action space $A$ is defined as differential drive command for the AVG, which is a 2-dimensional input $\vec{a}_t = [v_t, \omega_t]$ with $v_t \in [-1\,\text{m/s}, 1\,\text{m/s}]$ and $\omega_t \in [-1\,\text{rad/s}, 1\,\text{rad/s}]$. Here $v_t$ and $\omega_t$ are linear and angular target velocity inputs for the differential drive. The actions are carried out for 180 ms, resulting in an approximately 5.5 Hz operation frequency.

The reward function is formulated as:

$$r(t) = r_S(t) + \mathbb{1}_{CD} r_{CD}(t) + \mathbb{1}_C r_C(t) + \mathbb{1}_F r_F(t) + \mathbb{1}_G r_G(t),$$

where $r_S = -0.1$ represents a negative reward for each time step, $r_{CD} = -0.1$ denotes a small negative reward for collision with the dolly, and $r_C = -10$ corresponds to a large penalty for collision with non-dolly objects, i.e., walls or other obstacles. We set a small penalty for collision with dolly so as to encourage the agent to reach the proximities to the dolly. The term $r_G = 10$ is a positive reward when the forklift ends up with successfully reaching underneath the dolly. The last component $r_F = -0.05$ acts as a penalty of not driving forwards, i.e., when the velocity of the AVG is below 0.3 m/s. The symbol $\mathbb{1}_{CD}$, $\mathbb{1}_C$, $\mathbb{1}_F$ and $\mathbb{1}_G$ denotes the corresponding indicator function of whether that event happens. Note that our reward design does not require any map information, e.g., the distance between the dolly and the agent, so that it can be well applicable also to real-world training. Terminal state is reached once the Euclidean distance between the center of the dolly and the center of the robot is less than 0.3 m. Collisions with any objects will also result in an instant termination of the task.

To increase the generalizability of the learned policy, we inject domain randomization for each worker environment, e.g., shift of light sources, shape of cells, pattern of the floors and walls etc. The designed arena is shown in Figure 3. Particularly, we randomize the number of obstacles, the position of target dolly and initial pose of the AVG in the cell to avoid overfitting of the sensor readings on a single environment. The exact randomization scheme is demonstrated in Appendix C.

To accelerate training, we also implement proportional-based PER [55] as well as distributed RL, where we parallelize 9 agents for collecting the experience in different environments and a main training process. We followed an asynchronous update approach.

The worker thread sends the trajectory experience to the main training thread and gets an updated model copied from as long as it finishes an episode, the training thread is in charge of updating the actor and critic networks. The details of distributed version of SAC and its hyper-parameter setting are described in Appendix A.

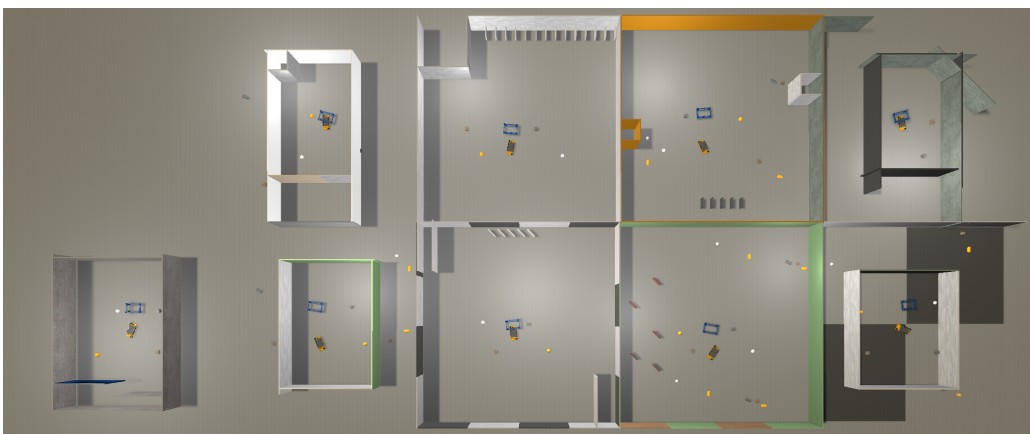

**Figure 3.** An Illustration of the designed training arena. It consists of 9 total cells of different sizes and layouts. For instance, the walls and floors feature different colors and patterns, the light sources differ also in each cell. The initial pose of robot, target dolly and the obstacles are also placed with some random settings. For details, please refer to Appendix C.

*3.4. Automatic Curriculum Learning: Extension of NavACL to NavACL-Q*

In this part, we introduce our improved Automatic Curriculum Learning approach *NavACL-Q* based on the original study *NavACL*. *NavACL* [20] is an Automatic Curriculum Learning method that specially addresses the challenges of robotic navigation. The idea of *NavACL* is to autonomously propose tasks of suitable difficulty to reduce overall training cycle and enhance the final performance. To automatically form a curriculum, *NavACL* uses a neural network $f_\pi$ to estimate the probability of the current policy $\pi$ solving task $l$, with $f_\pi^*(l) = 0$ for certain failure and $f_\pi^*(l) = 1$ for certain success. The success probability $f_\pi^*(l)$ is estimated based on a list of pre-defined task-specific geometric properties $l$ relevant of map knowledge, i.e., geodesic distance between goal and initial state, agent/goal clearance, relative initial angle and etc., altogether 5 properties.

The neural network $f_\pi$ is updated concurrently with the training of the policy $\pi$. For training, the network is provided with the input of $l$ from the most recent trained task and a binary label that indicates whether or not the respective task was solved successfully.

To determine which curriculum should be posed next, *NavACL* samples on *easy*, *frontier*, and *random* tasks with some defined probabilities. *Frontier* tasks refer to challenging situations and *random* tasks encourage exploration, while *easy task* prevents the catastrophic forgetting. Since agent's ability to solve the task changes dynamically during the training, the authors of *NavACL* used *adaptive filtering* (AF) as a criterion to evaluate in which category the generated candidate tasks fall into. Specifically, AF proposes a set of candidate tasks characterized by their respective geometric properties $L_c$ are forwarded through the network $f_\pi$. Then a normal distribution is fitted to the estimated success probabilities, which is formally expressed as $\mu_f, \sigma_f \leftarrow FitNormal(f_\pi^*(L_c))$. A task is regarded as *easy* if $f_\pi^*(l) > \mu_f + \beta\sigma_f$ and is classified as *frontier* when $\mu_f - \eta\sigma_f < f_\pi^*(l) < \mu_f + \eta\sigma_f$. The coefficients $\eta$ and $\beta$ are hyper-parameters.

Now we illustrate our improvements and extensions of *NavACL* algorithm to *NavACL-Q* algorithm. We adapt the 5 geometric properties of the network to our use case as shown in Table 2. Among them, we reduce the number of geometric properties by one, and combine our proposed idea. Rather than fully relying on the map properties, it is more favorable that the input features to success probability network can be more generalizable for tasks in different domains, e.g., navigation tasks, robotic manipulation tasks, games, etc.

In contrast, the current input features (geometric properties) need redesigning to fit other tasks. To cope with this, we propose using the initial Q-value from the critic network, as the learned Q-value should give a good estimate on how well the initial pose is, as shown in Section 3.1 . Thus, the Q-value is highly correlated with the success given the initial state. In this work, we append the estimated initial Q-value with other geometric properties together as the input to *NavACL*.

**Table 2.** The inputs for the success prediction network $f_\pi$ in *NavACL-Q*.

| | |
|---|---|
| Agent-Goal Distance | Euclidean distance from $s_0$ to $s_g$ |
| Agent Clearance | Distance from $s_0$ to the nearest obstacle |
| Goal Clearance | Distance from $s_g$ to the nearest obstacle |
| Relative Angle | The angle between the starting orientation and $\overrightarrow{s_0, s_g}$ |
| Initial $Q$-Value | The predicted $Q$-value $Q_\phi(s_0, a_0)$ from SAC critic network |

Moreover, we spot that AF could result in an undersampling on the *easy* tasks if $\mu_f + \beta\sigma_f > 1$. This could happen when either $\mu_f$ approaches 1 or $\sigma_f$ is large. The first case indicates that the agent reaches near-final performance and is thus of minor importance. Here we simply introduce an additional hyperparameter $\chi \in [0, 1)$, which acts as a threshold for *easy* tasks during the final stage of the training. The second corner case is handled by an additional condition that checks for the case that $\mu_f + \beta\sigma_f > 1$. If so, the *easy* condition is replaced with $f_\pi^*(l) > \mu_f$. All these modifications are shown in Algorithm 1.

---

**Algorithm 1:** GetDynamicTask-Q.

> **input** : Training timestep $t$; $f_\pi$; $\mu_f$; $\sigma_f$; Hyperparameters $\beta$, $\gamma$, $\chi$, $n_T$
> **output**: Task $l$

1  $taskType \leftarrow GetTasktype(t)$;
2  **for** $i = 0$ **to** $n_T$ **do**
3      $l \leftarrow RandomTask()$;
4      **switch** $taskType$ **do**
5          **case** *easy* **do**
6              **if** $\mu_f + \beta\sigma_f < 1$ **then**
7                  **if** $f_\pi^*(l) > \mu_f + \beta\sigma_f$ *or* $f_\pi^*(l) > \chi$ **then**
8                      **return** $l$;
9                  **end if**
10             **else**
11                 **if** $f_\pi^*(l) > \mu_f$ **then**
12                     **return** $l$;
13                 **end if**
14             **end if**
15         **end case**
16         **case** *frontier* **do**
17             **if** $\mu_f - \gamma\sigma_f < f_\pi^*(l) < \mu_f + \gamma\sigma_f$ **then**
18                 **return** $l$;
19             **end if**
20         **end case**
21         **case** *random* **do**
22             **return** $l$;
23         **end case**
24     **end switch**
25 **end for**
26 **return** $RandomTask()$

---

In the original work, they used proximal policy optimization [26], an on-policy DRL algorithm, whereas we implement SAC [34]. The advantage of SAC is mentioned in Section 3.1. We elaborate on the hyper-parameter of *NavACL-Q* in Appendix B.

### 3.5. Pre-Training of the Feature Extractor

We also investigate other alternatives to potentially increase the training speed besides automatic curriculum learning. One potential way is to pre-train the convolutional encoder in unsupervised learning manner, e.g., via auto encoders [66]. After pre-training, we initialize and fixed the weights of convolutional blocks shown in Figure 2 during the whole DRL training phase and the decoder are discarded. We examine whether a meaningful feature representation can facilitate the learning or not.

Here we demonstrate the details of pre-training the convolutional encoders in actor and critic network. The encoder structure is mentioned above. For decoder, we use symmetric architecture with transposed convolution [67] to increase feature map size. The output of the decoder has exactly the same shape as input with 4 channels, i.e., 4 consecutive frames. The loss function is defined as l2 pixel-loss between the reconstructed image and ground truth image averaged over all channels so that the underlying temporal relation between each frame is also reckoned with. The dataset consists of 50,000 interaction sequences from the agent's random interaction with the training environment.

## 4. Results

In this section, the training and testing results of our DRL approach on the navigation task are presented. We start by showing the training performance of the best variant in Section 4.1. For testing, we evaluate the learned policy systematically in an unseen environment featuring larger space with different layout and obstacles in Section 4.2. The testing also extrapolates the training scenarios in case of higher relative orientations between vehicle and dolly so that the robustness of the learned policy can be seen. In Section 4.3, a set of ablation studies are conducted, where the effects of our ACL approach *NavACL-Q* and a pre-trained feature extractor on the performance gain and training efficiency are investigated. We reveal their effects both in term of training and testing experiments. Specifically, *NavACL-Q* is compared with training with random starts and a pre-trained convolutional encoder is compared with a completely end-to-end training fashion. In Section 4.4, we compare all ablation variants to a map-based pipeline approach provided by Nvidia Isaac SDK [68] as a baseline approach to check whether our DRL agent outperforms the standard approach for navigation task.

### 4.1. Training Results

During the experimental phase, we investigate three variants for ablation studies. We name the variant that combines both *NavACL-Q* algorithm and pre-trained convolutional blocks as *NavACL-Q p.t.*, the variant with *NavACL-Q* algorithm but with end-to-end training (i.e., the convolutional encoder is also learned from scratch) as *NavACL-Q e.t.e.*, while the variant with pre-trained convolutional encoder but with random initial poses is abbreviated as *RND*. We append our abbreviation list also at the end of our script. A comprehensive comparison among three variants reveals how automatic curriculum learning and pre-trained feature extractor can facilitate learning process, which is presented in Section 4.3. In this section, we first demonstrate the training result of the best variant *NavACL-Q p.t.*

#### 4.1.1. Pre-Trained Convolutional Encoders

In this section, we are presenting the quality of the pre-trained convolutional blocks using auto encoder in Figure 4 with the training procedure mentioned in Section 3.5. After 100 epochs of training, the auto-encoder is able to reconstruct the image sequences with reasonable accuracy. We use these trained weights as initialization for actor and critic network of SAC and freeze them during training.

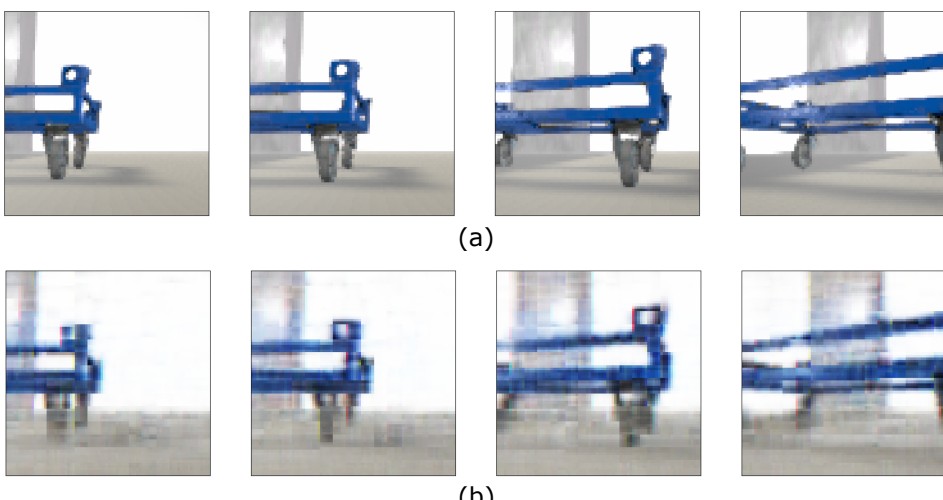

(a)

(b)

**Figure 4.** Comparison of (**a**) groundtruth image sequence and (**b**) reconstructed image sequence from auto-encoders.

### 4.1.2. Performance of *NavACL-Q* SAC with Pre-Trained Convolutional Encoders

We are presenting the learning curves of the best DRL variant during the complete training episodes in Figure 5. The algorithm is run a total number of three times, with each run consisting of 0.25 M episodes. This approximately corresponds to the wall clock time of 28 h and 5 M frames, where the training is split across three GPUs (Quadro RTX 8000 48 GB) with the CPU Intel Xeon Gold 5217. Figure 5 outlines the episodic return and success probability of reaching underneath the dolly. It can be observed that the final success probability approached 1, i.e., the mobile robot succeeds mostly in navigation from arbitrary defined initial position domain as shown in Appendix C. The episodic return also converges to the final value stably, which can be interpreted as converging to a near-optimal policy. The variance of episodic return and success probability is small at the end of training, signifying the learned policy registers similar patterns among three runs. The robustness of our algorithm is therefore evident. Moreover, our trained DRL agent further exhibits the adaptability to a non-stationary environment. We show that the agent is even able to reach the goal dolly that changes its location during the navigation process. For instance, the dolly is originally placed left front to the AVG, and the agent steers towards the goal direction. Then the target dolly is shifted from the left front of the agent to its right front, the AVG drives first backwards and adjusts its orientation towards the goal direction and then advances to the goal. The video illustrations of leaned policy are available in our supplementary materials. The generalizability of the trained agent serves as a great advantage of DRL and will be further discussed in Section 5.1.

### 4.2. Grid-Based Testing Scenarios

To exactly examine the robustness and the generalizability of the trained policy, we perform a systematic testing in an arena distinct from training. The test environment differs from the training environments in terms of shape, texture, and lighting conditions to enable the analysis of the methods generalization potential. We set the initial pose of the (2D location and orientation) in an exhaustive grid-based manner and checked the success probability of each initial position. The exhaustive testing features a 5 m × 5 m grid, partitioned into 0.5 m grid-cells, centered in front of the dolly. The grid thus represents 11 × 11 initial positions for the mobile robot. Figure 6 illustrates a schematic representation of the testing scenario. Furthermore, we test each initial positions with eight different initial robot orientations, given by the following list: $\{0°, \pm 45°, \pm 90°, \pm 135°, 180°\}$. For simplicity, the orientation of $0°$ can be approximately understood as the case where the robot is facing straight towards the goal and $180°$ corresponds to facing backwards from the goal. Please refer to Appendix C for details. The effect of partial observability aggregates with the

increasing angle. Each combination of position and rotation is tested nine times to obtain an averaged estimation of the success probability for each configuration (x, y, orientation), which corresponds to the total number of $11 \times 11 \times 8 \times 9 = 8712$ runs for each ablation variant. Figure 6 visualizes the test-scenario graphically.

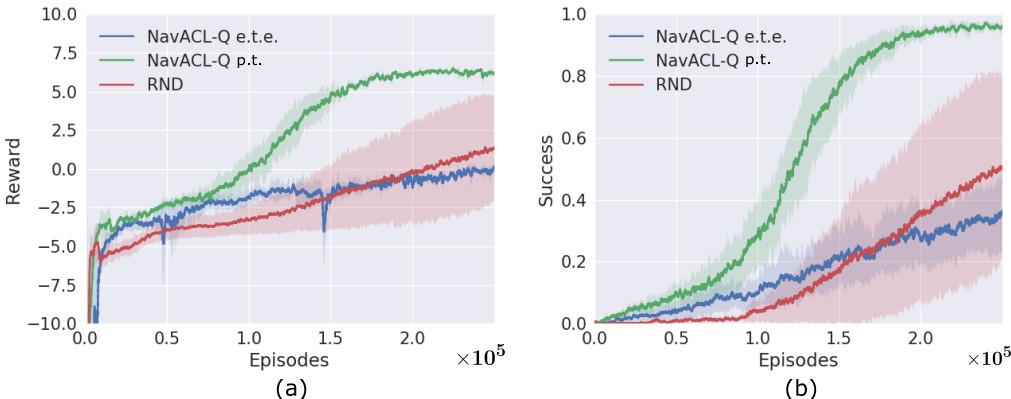

**Figure 5.** Learning curves of the three variants. (**a**) The episodic return, (**b**) The docking success probability per episode. These two statistics are presented as a moving-average over 500 episodes, where the solid line and shaded area illustrates, respectively, the mean and the variance over three runs.

It is noteworthy that the testing case features wider initial AVG orientation, lying between the interval of $[-180°, 180°]$, which extrapolates the defined one of $[-90°, 90°]$ in the training phase. In this way, one can also examine the performance of learned policy under the effect of partial observability.

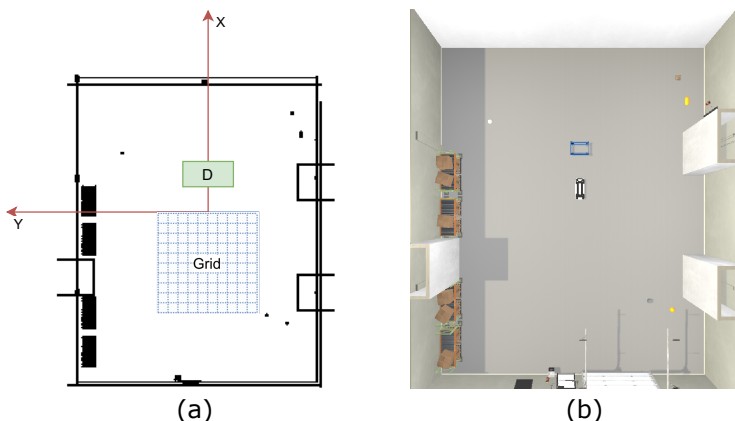

**Figure 6.** (**a**) Schematic representation of the grid-based test-scenario. The coordinate system to which the test-scenario refers is shown in red. The fixed dolly position is marked with "D". The blue grid represents the test zone divided into $11 \times 11$ positions. The grid and the dolly are scaled up for this illustration to improve visibility. (**b**) Graphical illustration of a $0°$ rotation test, conducted in the simulated testing-environment.

The best run among the three runs of *NavACL p.t.* is illustrated in Figure 7a. It is manifest that the agent scores approximately 100% probability to reach the dolly from a favorable initial starting position. The term 'favorable' stands for a relatively small initial robot orientation from the target dolly, e.g., $0°, \pm45°, -90°$. These cases suffer minimally from partial observation and the agent can reach goal mostly with proper turning and maneuvering. With aforementioned favorable initial rotation angle, the mobile agent mostly succeeds in navigating to the goal, except for left/right upper corner cases, i.e., the region near $-3$ m in x-axis and $-2$ m in y-axis for rotation angle $-90°$. This result is reasonable as such corner case is deemed as difficult start position as the robot cannot capture the target from the RGB camera and the mobile robot has to make a series of

adjusting maneuvers to reach upright underneath the dolly, similar to parking the cars to a narrow slot, hence resulting in a sub-optimal policy. Additionally, such corner cases are not sufficiently frequently sampled, resulting in a potential class imbalance problem. Consequentially, DRL algorithm fails to learn from these samples well.

With the increasing initial rotation angle of $\pm 135°$ and $180°$, which extrapolates the defined orientation domain of $[-90°, 90°]$ during training, the success probability drops significantly and the partial observability severely affects the performance. For the majority of failure cases, the mobile robot exhibits one of following behavior patterns: (i) Consistently making cycling movement, with some runs tending to gradually approach the target, but ending up with reaching maximal allowed steps. (ii) Going straight towards collision without moving forwards or backwards. (iii) Going towards an obstacle, but circling around in its proximity, exceeding maximal allowed steps. A potential reason for such failure cases is that the learned policy cannot generalize well to extrapolated tasks not seen in training, whereas the for intrapolated tasks ($\pm 45°$ and $0°$), the policy generalizes well.

### 4.3. Ablation Studies

In the above sections, we demonstrate the training and testing results of the best variant *NavACL-Q p.t.* In this section, we examine the effect of a pre-trained convolutional encoder and *NavACL-Q* on the learning performance, which, respectively, corresponds to two additional variants *RND* and *NavACL e.t.e.* We also run the remaining two variants in the exact setting as *NavACL-Q p.t.* also with three runs for each variant and demonstrate the complete training and testing results in Figures 5 and 7.

#### 4.3.1. Ablation Studies: Effects of Automatic Curriculum Learning

To investigate the effects of our automatic curriculum approach *NavACL-Q*, we compare it with the variant *RND*, where *RND* has the same setting as *NavACL-Q p.t.* except that it samples the initial state randomly from the defined boundary. It is first to be noted that *RND* is already an approach encouraging the exploration and alleviates the task difficulty compared to a fixed distant initial position from the target [15]. If a better training performance can be witnessed from *NavACL-Q*, then the effectiveness of our automatic curriculum learning approach can be verified. We demonstrate the comparison both in training and testing performance.

Figure 5 reveals that in the training phase the *RND* method imposes the highest training variance. One of the three trials meets the 90% threshold in the *RND* case, though the other two trials have not exceeded 40% performance, resulting in an average final performance of approximately 50%. In contrast, the variance of *NavACL-Q e.t.e.* remains small, i.e., more robust. This can also be validated from the variant *NavACL-Q e.t.e.*, which also incorporates the component of *NavACL-Q*. Moreover, the *NavACL-Q p.t.* exhibits a consistently faster improvement at the initial stage of training. With $1M$ steps, *RND* just starts to show a sign of improving in terms of success rate whereas *NavACL-Q p.t.* has already reached the average success rate of 30% in Figure 5b. From these observations, it can be concluded that *NavACL-Q* indeed facilitates the training in terms of both final converged value and rate of convergence.

We further present a summary statistics for a comparison on the testing performance in Table 3, *NavACL-Q p.t.* achieves approximately 90% success rate in navigating underneath the dolly among four out of five intrapolated initial orientations, outperforming *RND* by at least 30%. Moreover, *NavACL-Q p.t.* reveals the best performance averaged over all orientations, surpassing *RND* by 11% for all tested orientations including the extrapolated ones. Overall speaking, *NavACL-Q p.t.* converges to a better policy than *RND* in the testing case. Moreover, both *RND* and *NavACL-Q p.t.* suggest general decreasing tendencies of success rates with increasing relative initial orientations, which is reasonable as the effect of partial observability aggregates with higher relative orientations.

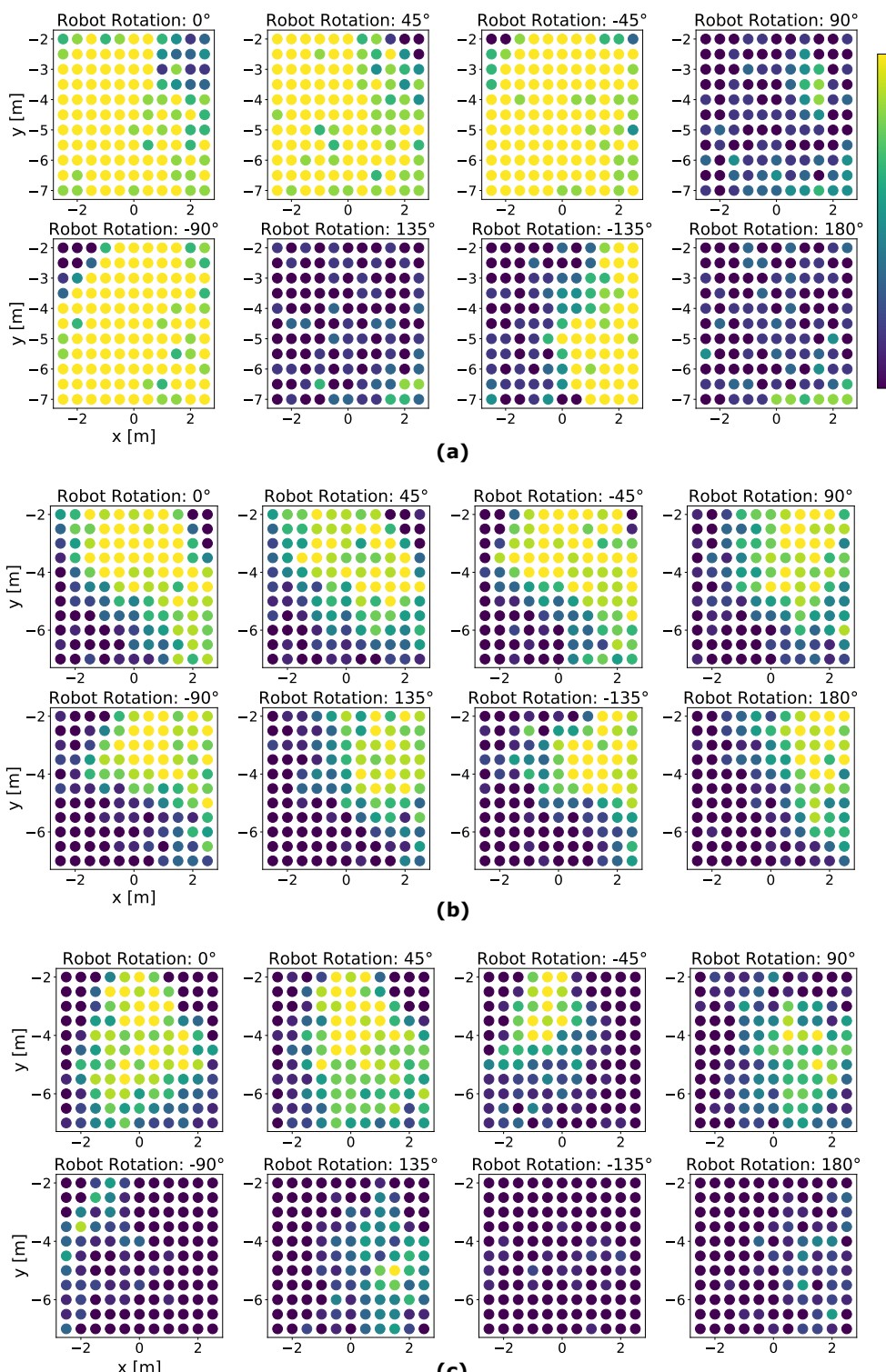

**Figure 7.** Color-coded illustration on the grid-based testing result of one fully trained *NavACL-Q* agent. The average performance for each position on the grid is represented by a colorized circle, where yellow color indicates a high success rate and blue color indicates near-zero success probability. (**a**) The testing result of *NavACL-Q p.t.* (**b**) The testing result of *RND* (**c**) The testing result of *NavACL-Q e.t.e.* A further summary of the statistics is available in Table 3.

**Table 3.** The statistics of testing results are presented. The averaged success rate of reaching the target for each ablation variant and baseline approach are shown. The averaged success rate is calculated as the mean success rates over $11 \times 11$ grid points from Figure 7.

| Relative Orientation of AVG to Target | Average Success Rate | | | |
|---|---|---|---|---|
| | *NavACL p.t.* | *RND* | *NavACL e.t.e.* | **Baseline** |
| 0° | **86.6%** | 58.5% | 50.3% | 16.5% |
| −45° | **93.7%** | 55.6% | 25.5% | 3.3% |
| +45° | **88.0%** | 52.2% | 53.5% | 5.0% |
| −90° | **90.5%** | 43.2% | 8.9% | 0% |
| +90° | 18.5% | **45.5%** | 32.0% | 0% |
| −135° | **48.2%** | 36.9% | 2.2% | 0% |
| +135° | 11.9% | **37.1%** | 19.8% | 0% |
| +180° | 15.7% | **34.2%** | 8.0% | 0% |
| Mean of $\{0°, \pm45°, \pm90°\}$ (Intrapolated Tasks) | **75.5%** | 51.1% | 34.1% | 5.0% |
| Mean of $\{\pm135°, 180°\}$ (Extrapolated Tasks) | 25.3% | **36.1%** | 10.0% | 0% |
| Mean of All Orientations | **56.6%** | 45.4% | 25.0% | 3.1% |

We take a closer look at whether the success prediction network $f_\pi$ in *NavACL-Q p.t.* shows meaningful predicted success probability and how it evolves with the training stages. To examine this effect, the outputs of $f_\pi$ are evaluated across different stages of training from one of the three runs. Figure 8 illustrates predicted task success rate in the defined regions with respect to two geometric properties, initial robot orientation and the Euclidean distance between initial robot pose and target dolly. These two properties give a straightforward view on the task difficulty. For instance, a small initial rotation angle with close distance is regarded as optimistic initial position. It is hypothesized that a well-learned $f_\pi$ should show a increasing tendency on success probability as the training progresses. Besides, the prediction network should also distinguish favorable initial poses from unfavorable ones.

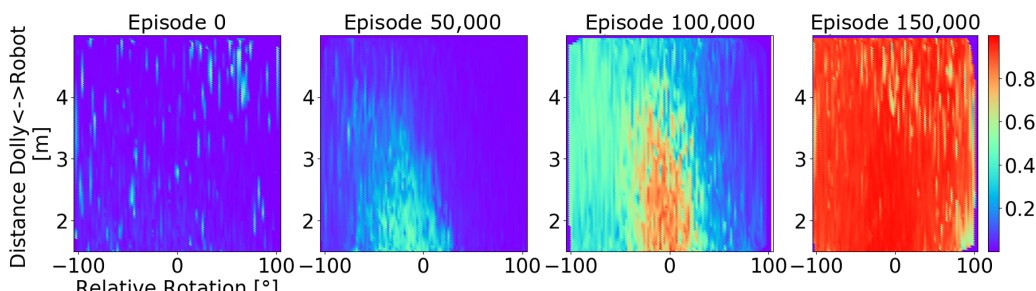

**Figure 8.** Two-dimensional interpolation of the success probability estimated by $f_\pi$ at different stages of training, where red areas indicate high success probability estimates and blue areas indicate low success probability estimates. In this case, the plot is generated across the geometric properties *Agent-Goal Distance* and *Relative Angle*. The individual plots consist of the success predictions of the 10,000 tasks that followed the displayed episode.

As can be observed from Figure 8, at the initial training stage, e.g., episode 0, the agent behaves totally in a random fashion and a general low predicted success value can be expected. With the training progressing, e.g., episodes 50,000 and 100,000, the prediction network suggests an increment in success rate in the regions of favorable initial poses, where the increment also spreads to non-favorable initial pose. In both cases, tasks with

low relative rotation and distances less than 4 m exhibits a significantly higher success probability. Towards the end of training, the entire task space reaches an estimated success probability close to 100%.

We further inspect task distribution from the curriculum in different training stages. Figure 9 displays a set of histograms, which accounts for the number of tasks in terms of one geometric property *Agent-Goal Distance*. In the first 10,000 episodes, a random pattern with respect to the *Agent-Goal Distance* for the agent *NavACL p.t.* is present. This is reasonable as the agent behaves more randomly at the beginning and mostly ends up with not reaching the target. With merely negative experience, the success prediction network cannot distinguish *easy* task from *frontier* task, hence reaching a more or less random pattern. In the intermediate stage, represented by episodes 50,000:60,000, where the agent starts to learn in the initial positions with small relative distance but still fails in large initial distance. This can additionally be verified in Figure 8. In this phase, *easy* and *frontier* task corresponds to the regions with close distances. This is referred to as a more concentrated distribution that can be found with the featured goal distances in the range of 1.5 m to 1.5 m. Note that the definition of *easy* and *frontier* task also evolves with the training stage. In a later training stage, represented by episodes 150,000:160,000, the success prediction network $f_\pi$ has mostly successful predictions covering all the relative distances in Figure 8. In this case, large initial distances can also be classified as *easy* task or *frontier* one, resulting in a random sampling. This tendency is in accordance with the anticipated behavior of *NavACL-Q*. The *RND* agent on the other hand, is trained based on randomly sampled tasks only, therefore it shows a uniform distribution across the defined distance domain.

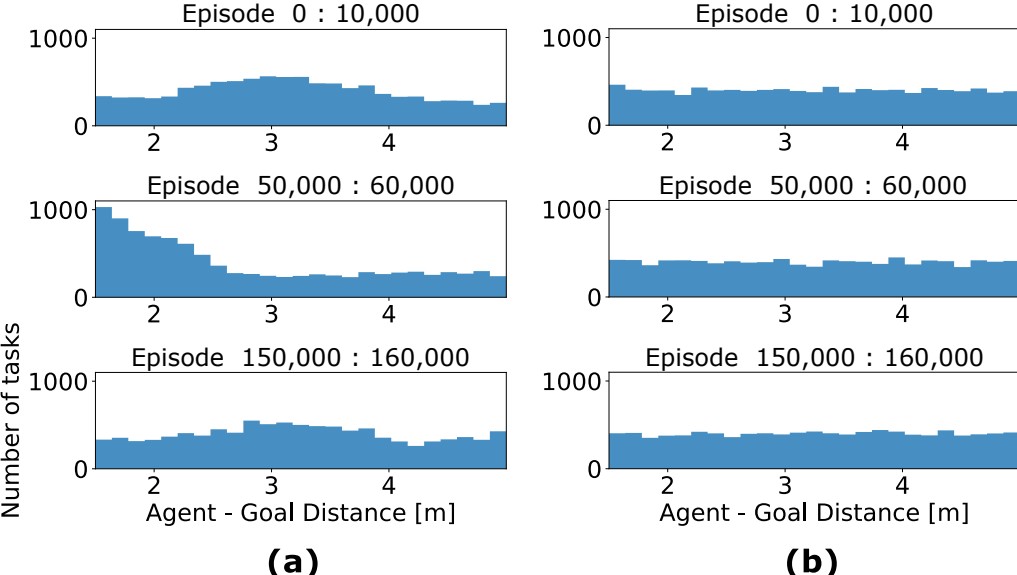

**Figure 9.** Comparison of the task selection histograms with respect to the *Agent-Goal Distance* geometric property of exemplary training outcomes. We have recorded the statistics of initial position in terms of *Agent-Goal Distance* among different training stages, each with 10,000 episodes. The histogram counts the corresponding number of initial states in the defined distance bins among each 10,000 episodes. (**a**) represents task distribution of a *NavACL-Q* agent and (**b**) illustrates the task distribution of an *RND* agent.

### 4.3.2. Ablation Studies: Effects of Pre-Trained Convolutional Encoder

For the training performance, Figure 5 demonstrates that *NavACL-Q* and *NavACL-Q e.t.e.* perform similarly in terms of return and success rate during the first quarter of the training stage. Intermediate training performance with an approximately 30% success rate is reliably achieved by both approaches. Nevertheless, robust policies with final performance over 90% are exclusively learned by agents that utilize the pre-trained feature extractor. *NavACL-Q e.t.e.* attain similar variances as *NavACL-Q*, but the final performance

stagnates around 30%. In an additional experiment, the maximum number of episodes is increased to 0.4 M, yet still no robust policy with success rates above 60% can be achieved in the *NavACL-Q e.t.e.* case.

In the grid test case, *NavACL-Q e.t.e.* also ends up with overall worse performance than *NavACL-Q p.t.* and *RND*. Table 3 demonstrates that *NavACL-Q e.t.e* achieves 41% lower success rate in navigation than *NavACL-Q p.t.* among all intrapolated tasks and 31% lower for all tested orientations. The most successful navigation trials happen when the initial robot rotation is 0°, corresponding to the easiest scenario. As the rotation angles increases, the successful probability drops quickly. With such comparison, it is obvious that the pre-trained network greatly boosts the performance as well as reduces the overall computation expense. We discuss this effect in Section 5.2. Interestingly, it can be also observed that the performance gain of a pre-trained convolutional encoder is more significant than how *NavACL-Q* boosts the performance, especially in the test case, as Table 3 demonstrates that *NavACL-Q e.t.e* exhibits 16.4% lower success rate than *RND* among all intrapolated tasks and 20% lower rate averaged for all orientations.

*4.4. Comparison to a Map-Based Navigation Approach*

We further compare the result of our learning approach to a full perception and navigation pipeline provided by NVIDIA Isaac SDK™ [21], which is deemed as a baseline approach. This baseline application is specifically designed for the load carrier docking task. In contrast to our solution, the baseline uses a global map for multi-LiDAR Localization of the robot. The target pose for the robot is determined by object detection followed by 3D pose estimation of the dolly. In the baseline application the camera resolution is $720 \times 1280$ and the number of the LiDAR beams used for localization is 577 per sensor, which have a maximum detectable range of 20 m. The used mobile robot except for the camera and the LiDAR resolution, remains the same.

To find the goal position for the robot, the pose of the dolly is inferred from the frontal facing camera of the robot. This poses a constraint that the dolly has to be detected in the input image. For the baseline approach, this is done by using DetectNetv2 [69], which was pre-trained on real images and then fine-tuned for dolly detection by using randomized images of the dolly, created with IsaacSim Unity3D [21]. DetectNetv2 generates a 2D axis-aligned bounding box for a detected dolly. The detected bounding box is used to create a cropped image of the dolly, which forms the input into a pose estimation model. In this case, the pose estimation model is based on PoseCNN [70]. The output of the PoseCNN network is an estimated orientation and translation of the dolly. Given an image of the dolly, the perception pipeline estimates the 3D pose of the dolly. This pose is transformed into the global coordinate-frame and serves as a target pose. Then, a local planner based on the linear quadratic regulator navigates the robot under the dolly.

We also conduct the same grid testing as mentioned in Section 4.2 for the baseline approach. Since the baseline approach enforces that the target dolly must be captured with a RGB camera, it is only possible to show the result with the initial orientation angle of 0° and ±45°. The baseline method achieves the success rate of 100% in the 0° orientation case under the condition that the displacement on x-axis remains below 1 m, and the distance in y-direction remains below 5 m, which is illustrated in Figure 10. However, the baseline approach proves incapable of performing the docking maneuver once the distance on the x- or y-axis surpasses the mentioned limitations. In the ±45° case, only few positions are solved successfully, all of which provide full visibility of the dolly. The ±45° cases require y-displacements between −2 m and −4 m for successful docking maneuvers. From Table 3, it can also be witnessed that the baseline approach achieves overall much worse performance compared to all other DRL variants in all initial orientations. As a conclusion, our learning approach definitely outperforms the baseline with larger initial orientation and distances to the target.

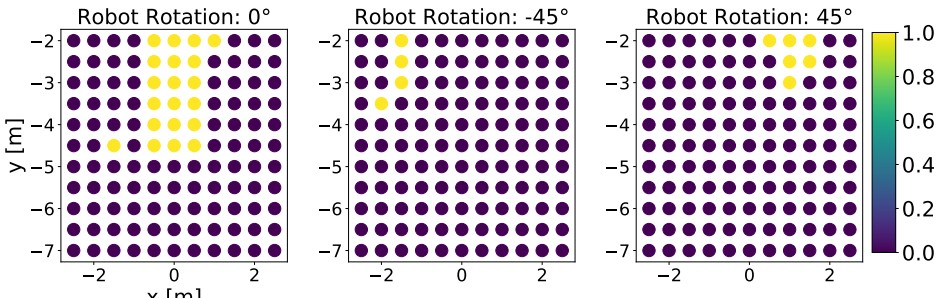

**Figure 10.** Similar to Figure 7, we demonstrate a color-coded illustration of the grid-based testing result of the baseline approach. The yellow color indicates a high success rate and blue color indicates near-zero success probability.

## 5. Discussion

In this section, we discuss the pros and cons of our learning approach compared to the map-based baseline approach in Section 5.1. Additionally, we provide some intuitive learned trajectories of our ablation variants and baseline approach for a qualitative description. In Section 5.2, we interpret the results from ablation studies as well as its correlation to other related works. For Sections 5.3 and 5.4, the result of intermediate trials and potential improvements of *NavACL-Q* approach as future work are discussed.

### 5.1. Learned Behavior of the Agent

As mentioned in Section 4.1.2, one major advantage of a successfully-trained DRL agent on the navigation task is the adaptability to a non-stationary environment. The mapping from raw sensory observation to the action fully relies on the learned DNN. As a comparison, map-based approaches require updating of a map and perform re-planning, which causes additional computation overhead and suffers from potential error of an inaccurate map. The corrective maneuver of DRL agent is naturally acquired from the experience encountered during training given sufficient exploration, i.e., learning from trial and error.

In addition, we are illustrating the learned trajectories of the map-based baseline approach and DRL variants. For a fair comparison, three scenarios have been chosen in a way that the baseline is able to perform the navigation successfully, i.e., dolly visibility in the frontal RGB camera is given. Each of subplots in Figure 11 illustrates one scenario with the leaned trajectories from *NavACL-Q p.t.*, *NavACL-Q e.t.e.*, *RND* and baseline given the same initial and goal state.

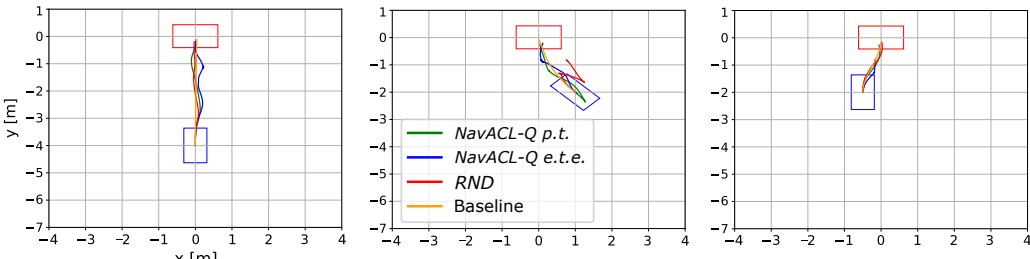

**Figure 11.** A selection of driven trajectories from three different initial positions, where the orange line represents the baseline trajectory, the blue line represents the *NavACL-Q e.t.e.* trajectory, the green line illustrates the *NavACL-Q* trajectory and the red line depicts the trajectory of the *RND* case. Some clipped trajectories signifies that the agent ended up with collision.

The quantitative illustration shows that the map-based approach provided by NVIDIA Isaac SDK returns optimal trajectory with minimal maneuvers in orientation adjustments, while DRL agents exhibit near-optimal ones. This sinusoidal behavior is natural as the DNNs can hardly eliminate approximation errors to 0. Additionally, SAC encourages the

agent to show stochastic behavior by maximizing the policy entropy and the illustrated trajectory is one sample from the learned policy distribution.

Our intermediate trials also reveals that a good randomization of the training is essential to a more generalized policy. In our setting, the position of the goal dolly needs to be sufficiently randomized in the cells. In some intermediate trials, where the degree of the randomization of the goal dolly is limited, e.g., more concentrated on one area in the cell, the ultimately learned agent tends to merely reach the defined target regions in training, but not towards the true target position in testing. This interesting phenomenon indicates that the robot is sometime more reliant on the LiDAR readings to infer the goal position instead of visual observation. Therefore, a good randomization of the target's position in the arena helps prevent the agent from relying merely on the LiDAR distance reading to infer the goal position.

We have also tested the agent's ability to navigate from very far distance to the goal, i.e., larger than 10 m. It is first spotted that the agent also makes cycling movements with a tendency of approaching the target, when the agent is within the distance of roughly 4 m away. Then it ceases the cycling motion and behaves near-optimally towards the goal. This motion pattern can be interpreted as having not yet learned to reach the goal positions in extrapolated task with larger initial distance than in the training task. Hence, our hypothesis for further improving the generalizability of learned policy is to equate the training domain to the target domain, otherwise the DRL is likely to exhibit very limited performance in extrapolated tasks.

### 5.2. Effects of Pre-Trained Feature Extractor

We have already seen that the variant *NavACL-Q p.t.* definitely outperforms *NavACL-Q e.t.e.* Such findings are also consistent with other research work. In [71], the car (agent) learns to drive in the street rationally with frontal cameras and it is expected to stop at the red light. They pre-train a feature extraction layer similar to an auto-encoder version with additional loss on the traffic lights, so that the information from traffic lights is accentuated. Hence, the car has learned to react correctly to the traffic light signal. They report a significantly better training efficiency and converged performance with a pre-trained semantic segmentation feature extraction layer than learning from scratch. The work of [72] further explores the possibility of performance enhancement when decoupling feature extraction from policy learning. They propose learning meaningful feature extraction via considering inverse dynamics $p(a_t|s_t, s_{t+1})$, reward prediction $p(r_{t+1}|s_t, a_t)$ and reconstruction from visual observations. In their ablation studies, the variants of auto-encoder, random feature extraction and end-to-end learning are also compared jointly. The result shows that there is no variant dominating other pre-trained feature extractor, but with pre-trained feature extraction, it generally outperforms end-to-end learning. This is also similar to the findings of [73], where the agent behaves better with a set of pre-trained feature extractors than its counterpart. In addition, different sets of pre-trained feature extractors are beneficial to different purposes, e.g., exploring the environment or reaching the goal state. With a pre-trained network, the overall training time is also greatly reduced, however, at a cost of being no more end-to-end learning.

### 5.3. Potential Improvements on NavACL-Q

We have indeed verified the effectiveness of our automatic curriculum learning approach *NavACL-Q* in Section 4.3.1. Nonetheless, we still spot some cases where *NavACL-Q* may fail despite the improvements on the original *NavACL* algorithm. This happens when the agent fails even in the initial optimistic regions quite often, ending up with much more negative samples than positive ones. As a result, *NavACL* cannot distinguish *easy* tasks from *frontier* ones or *random* tasks due to severe class imbalance problems. Under these circumstances, the initial poses with low success probability are interpreted as *easy*. Consequentially, the curriculum fails to propose beneficial intermediate tasks, and is behaving similarly to random starts.

One potential solution to this issue is to start curriculum proposing when the agent performs sufficiently well on the favorable initial state. With guaranteed success on a favorable initial task, the *NavACL* algorithm can distinguish *easy* task from *frontier* task. This idea will be investigated in our future work.

The other improvement is the generalization of *NavACL-Q* to be domain independent. The current input of the success prediction network $f_\pi$ still considers the domain dependent properties such as distance to goal and initial rotation, which requires additional manual design. A more meaningful approach is to leave out domain-dependent handcrafted features and to retain only domain-independent ones, for instance, $Q$-value of the initial state-action pair, which will be used in most DRL algorithms. Such settings will easily generalize *NavACL-Q* to other tasks, which is worth future investigation.

### 5.4. Effects of Problem Formulations on the Performance

To address the partial observability, we have taken a simple approach by stacking 3 previous frames along the channel dimension according to [19]. However, the trained agent still shows limited performance with larger initial rotation angle away from the target dolly. One potential explanation is that the historical observation is still not long enough to mitigate partial observability. In the work of [49], they use LSTM with increasing the historical length of 20 steps. The performance is reported to be better than stacking 3 previous frames. This approach be potentially effective, but at the cost of a longer training duration.

We have also tried a simpler environmental setting, where the agent tries to navigate to the door and the mobile robot is only equipped with frontal grey-scaled camera and three previous frames are stacked. Importantly, the door is designed with distinct grey-scaled color from other objects so that the agent can recognize the target state from the grey-scaled observation. We have merely implemented the SAC algorithm with random starts, but without pre-trained feature encoders. Interestingly, the learned policy is mostly optimal and the agent learns to rotate at the beginning to search for the goal and moves towards the door as soon as it gets detected. An investigation of whether increasing historical length can significantly increase the performance with larger initial rotation angles will be investigated further.

In some intermediate trials, we have found that the agent relies on the LiDAR readings to infer the goal position instead of relying on the visual observations. Therefore, we have tried one variant forcing our RL agent to perform goal detection via visual observation, whereas LiDAR readings are only intended for collision avoidance. To this end, the maximal detectable range of LiDARs has been set to a maximum of 1.5 m, and the pre-processing of the LiDAR observation is also rescaled into $[0, 1]$ correspondingly. Strangely, only with this change, the complete training ends up with failure. The agent fails highly frequently even with the simplest optimistic initial state. The reason for this is still unknown, might be potentially related with network initialization as mostly of beams ends up with the maximal readings of one after rescaling, breaking the assumption that the input features a normal distribution on which most weight initialization is based.

We have also conducted some simple trials for reward shaping. In one attempt, the reward term $r_{CD}$ has been set to be of the same as $r_C$, namely, not distinguishing collision with dolly or other obstacles. Our findings show that the small penalty of dolly collision definitely encourages the agent to approach that area and simplifies the training.

### 6. Conclusions

In this paper, we have demonstrated an approach of deep reinforcement learning with automatic curriculum learning on solving challenging navigation tasks with LiDAR and frontal-view visual sensors in intralogistics. The key challenge of task lies in a DRL problem formulation to deal with sparse positive experience, multi-modal sensory perception with partial observability, long training cycles and the need for accurate maneuvering. To address these problems, distributed soft actor-critic with *NavACL-Q* algorithm haven

been proposed. Our learning approach is completely mapless (no efforts for mapping and localization) and without human demonstration and relies entirely on the power of neural networks to directly map multi-modal sensory readouts to the continuous steering command. In addition, the reward formulation has been designed in a manner that can be directly used in real case.

The results show that our DRL-agent has a significantly better performance than the baseline map-based navigation approach provided by Nvidia. The baseline approach is only applicable to the case where the frontal RGB camera captures the target dolly and is merely within 3 m distance from the goal. In contrast, our DRL-agent has managed navigation task from up to 3 m further distances and up to 45° higher relative angles compared to the baseline approach. In testing case, our learning approach achieves the task with average success rate for different initial robot orientations, outperforming the baseline approach by 53%. Furthermore, our ablation studies reveal that our automatic curriculum learning approach *NavACL-Q* facilitates the learning efficiency compared to random starts with a final performance gain of roughly 40% in training and 11% in testing on average, and a pre-trained feature extractor boosts final training and testing performance by approximately 60% and 31%, respectively.

## 7. Patents

This paper is a part of one running patent.

**Supplementary Materials:** The following are available at https://github.com/ai-lab-science/Deep-Reinforcement-Learning-for-mapless-navigation-in-intralogistics, accessed on 30 January 2022.

**Author Contributions:** Conceptualization, B.H., B.A., M.B., H.X.; methodology, H.X., B.H.; software, B.H.; formal analysis, H.X., B.H.; investigation, B.H., H.X., B.A. and M.B.; resources, B.A., M.B.; data curation, B.H. and H.X.; writing—original draft preparation, H.X., B.H.; writing—review and editing, H.X., G.S., E.R., B.H. and B.A.; visualization, H.X., B.H.; supervision, B.A., M.B., E.R., G.S. All authors have read and agreed to the published version of the manuscript.

**Funding:** This research is funded by the Deutsche Forschungsgemeinschaft (DFG, German Research Foundation)–No 430054590 (TRAIN, to E.R.). The work is done in KION Group AG, Technology and Innovation.

**Institutional Review Board Statement:** Not applicable.

**Informed Consent Statement:** Not applicable.

**Data Availability Statement:** The data presented in this study are openly available in https://github.com/ai-lab-science/Deep-Reinforcement-Learning-for-mapless-navigation-in-intralogistics, accessed on 30 January 2022.

**Conflicts of Interest:** The authors declare no conflict of interest.

## Abbreviations

The following abbreviations are used in this manuscript:

| | |
|---|---|
| AVG | Automated Guided Vehicle |
| RL | Reinforcement Learning |
| DRL | Deep Reinforcement Learning |
| ACL | Automatic Curriculum Learning |
| SLAM | Simultaneous Localization and Mapping |
| SAC | Soft Actor-Critic |
| *NavACL-Q p.t.* | NavACL-Q with pre-trained convolutional encoder using Soft Actor-Critic |
| *NavACL-Q e.t.e* | NavACL-Q with Soft Actor-Critic, end-to-end learning |
| RND | Soft Actor-Critic with pre-trained convolutional encoder using random starts |
| PER | Prioritized Experience Replay |
| DQN | Deep Q-Network |
| LSTM | Long Short-Term Memory |

## Appendix A. Details for Training Via Soft Actor-Critic

In this part, we elaborate on the settings of distributed SAC as our DRL algorithm.

The network architecture of actor and critic is firstly shown. Figure 2 has already demonstrated an overall network design, and the detailed structure of convolutional blocks is illustrated in Figure A1. We also perform zero-padding for all previous rewards and actions $O_{ar}$ when the current time horizon is smaller than the defined historical length, i.e., four in our case. For the visual observation $O_v$, we perform replicate paddings of the RGB image at $t = 0$.

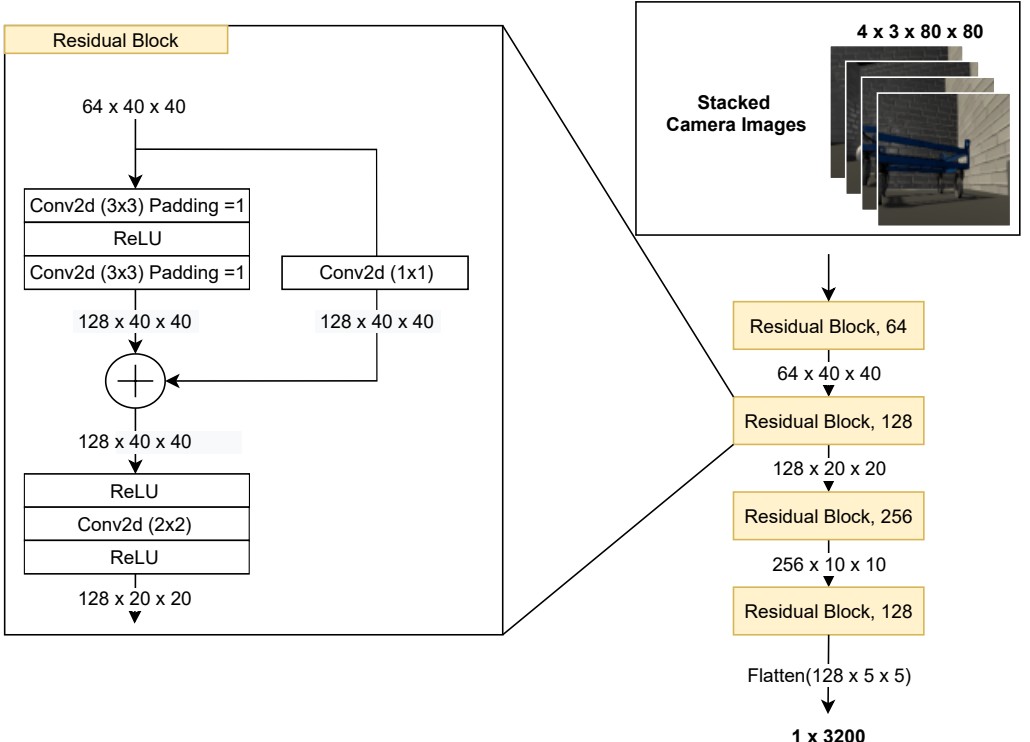

**Figure A1.** Illustration of encoder part for the stacked camera images. Four Residual blocks are used. In the left panel, the architecture of the residual blocks is illustrated. The first two convolutions use $(3 \times 3)$ filters, then the identity is concatenated to the output of the first two convolutions. Finally, we down-sample the image by half using a convolution with a filter of $(2 \times 2)$ and a stride of 2 according to [74].

As described in Section 3.3, we accelerate the training speed and improve the generalizability of the agent by paralleling nine agents, resulting in a distributed version of SAC. We show the pseudo code in Algorithms A1 and A2 for both worker process and master process. The asynchronous method for experience gathering is analogous to asynchronous advantage actor-critic [25]. Each worker process gets a copy of the shared actor from the main process and collects experience asynchronously. After one worker has completed an episode, the gathered experience is stored in a shared episode replay queue, and a new version of the shared policy is obtained from the master process. Concurrently, the master process updates the actor and critic networks and gathers all workers' experience from the shared episode replay queue to fill a PER buffer. The hyper-parameter settings are demonstrated in Table A1.

---

**Algorithm A1:** Distributed Soft Actor-Critic—Worker Process.

**input** :$\phi, \overline{\theta}, f_\pi, \mathcal{E}_s, env$ ;     ▷ Shared parameters for the policy, the Q-Function
and the `NavACL-Q` network, shared episode replay queue, and a target
environment for interaction

**input** :$L$ ;        ▷ A task database consisting of initial states based on *env*

**1 while** *True* **do**

**2**    $\mathcal{E} \leftarrow \varnothing$ ;                ▷ Initialize an empty local episode replay buffer

**3**    $\overline{\phi} \leftarrow \phi$ ;        ▷ Create a local policy network copy $\overline{\phi}$ for the next episode

**4**    $L_r \leftarrow$ randomly sample 100 tasks from $L$ $\mu_f, \sigma_f \leftarrow FitNormal(f_\pi^*(H_r))$
$task = GetDynamikTask - Q(\overline{\theta}, \mu_f, \sigma_f, H)$ ;     ▷ Use Nav-ACL-Q to get a task
that fits the current ability of $\phi$

**5**    **while** *maximal episodic length not reached* **do**

**6**      $a_t \sim \pi_{\overline{\phi}}(a_t|s_t)$ ;         ▷ Sample an action according to the local policy

**7**      $s_{t+1} \sim p(s_{t+1}|s_t, a_t)$ ;            ▷ Sample transition from the environment

**8**      $\mathcal{E} \leftarrow \mathcal{E} \cup \{(s_t, a_t, r_t, s_{t+1})\}$ ; ▷ Store the transition in the local episode
replay buffer

**9**    **end while**

**10**    $\mathcal{E}_s \leftarrow \mathcal{E}_s \cup \mathcal{E}$ ; ▷ Append the locally recorded episode to the shared episode
replay queue

**11 end while**

---

**Table A1.** Hyperparameter settings of SAC algorithm.

| Distributed Soft Actor-Critic Hyperparameters | |
| --- | --- |
| **Parameter** | **Value** |
| Discount factor $\gamma$ | 0.999 |
| Target smoothing coefficient $\tau$ | 1 (hard update) |
| Target network update interval $\eta$ | 1000 |
| Initial temperature coefficient $\alpha_0$ | 0.2 |
| Learning rates for network optimizer $\lambda_Q, \lambda_\alpha, \lambda_\pi$ | $2 \times 10^{-4}$ |
| Optimizer | Adam |
| Replay buffer capacity | $2^{20}$ (Binary Tree) |
| (PER) prioritization parameter $c$ | 0.6 |
| (PER) initial prioritization weight $b_0$ | 0.4 |
| (PER) final prioritization weight $b_1$ | 0.6 |

---

**Algorithm A2:** Distributed Soft Actor-Critic—Master Process.

**input** : $\theta_1, \theta_2, \phi, Env, b, m$ ; ▷ List of environments and the batch sizes for the SAC and the NavACL updates

1　$\overline{\theta_1} \leftarrow \theta_1, \overline{\theta_2} \leftarrow \theta_2$ ; 　　　　　　　　　　　▷ Initialize target network weights
2　$\mathcal{D} \leftarrow \varnothing$ ; 　　　　　　　　　　　　　▷ Initialize an empty PER replay buffer
3　$\mathcal{E}_s \leftarrow \varnothing$ ; 　　　　　　　　　　▷ Initialize an empty, shared episode replay queue
4　$\mathcal{L}_m \leftarrow \varnothing$ ; 　　　　　　　　　　　▷ Initialize an empty task result set
5　$init(f_\pi)$ ; 　　　　　　　　　　　▷ Initialize the *NavACL-Q* network weights
6　$n\_updates \leftarrow 0$ ; 　　　　　　　　　　　　　▷ Number of SAC updates
7　**for** *agent_index* $\leftarrow 0$ **to** *num_agents* **do**
8　　　**Spawn Process**
9　　　│　AsynchronousExperienceGathering($\phi, \overline{\theta}, f_\pi, \mathcal{E}_s, env = Env[agent\_index]$)
10　　**end**
11　**end for**
12　**while** *True* **do**
13　　**for** *agent_index* $\leftarrow 0$ **to** *num_agents* **do**
14　　　│　**if** $\mathcal{E}_s \neq \varnothing$ **then**
15　　　│　│　$Ep \leftarrow \mathcal{E}_s.pop()$ ; 　　　　　　　　▷ Pop one episode from $\mathcal{E}_s$
16　　　│　│　$\mathcal{D} \leftarrow \mathcal{D} \cup Ep$ ; 　　　　　▷ Append the episode to the PER buffer
17　　　│　│　$\mathcal{L}_m \leftarrow \mathcal{T}_m \cup Ep_l$ ; 　▷ Append the task of $Ep$ and the result of $Ep$ to $\mathcal{L}_m$
18　　　│　**end if**
19　　　│　**if** $\mathcal{L}_m$ *contains m tasks and task-results* **then**
20　　　│　│　$f_\pi \leftarrow Train(f_\pi, \mathcal{L}_m)$ ; 　　　　　　　　　　　　▷ Train $f_\pi$
21　　　│　│　$\mathcal{L}_m \leftarrow \varnothing$
22　　　│　**end if**
23　　**end for**
24　　$\mathcal{B}, w_i \leftarrow sample(\mathcal{D}, b)$ ; 　　▷ Sample a batch of interactions from the PER replay buffer
25　　**for** *each iteration* **in** $\mathcal{B}$ **do**
26　　　│　**for** *each gradient step* **do**
27　　　│　│　$\theta_i \leftarrow \theta_i - \lambda_Q \hat{\nabla}_{\theta_i} w_i J_Q(\theta_i)$ for $i \in \{1, 2\}$ ; 　▷ Update the Q-function parameters
28　　　│　│　$\phi \leftarrow \phi - \lambda_\pi \hat{\nabla}_\phi J_\pi(\phi)$ ; 　　　　　▷ Update policy weights
29　　　│　│　$\alpha \leftarrow \alpha - \lambda \hat{\nabla}_\alpha J(\alpha)$ ; 　　　　　　　▷ Adjust Temperature
30　　　│　**end for**
31　　**end for**
32　　$n\_updates \leftarrow n\_updates + 1$;
33　　**if** $n\_updates \% \eta = 0$ **then**
34　　　│　$\overline{\theta_i} \leftarrow \theta_i$ for $\{1, 2\}$ ; 　　　　　　　▷ Hard Update since $\tau = 1$
35　　**end if**
36　**end while**

---

## Appendix B. Details for Training the *NavACL-Q* Algorithm

Here, we show the hyper-parameters of *NavACL-Q* algorithm. The success prediction network $f_\pi$ consists of two dense hidden layers with 32 nodes each and the ReLU [75] as non-linear activation function. We use a sigmoid function for the output layer to limit the output-range to $[0, 1]$ together with binary entropy loss. Relevant details are listed in Table A2.

**Table A2.** Hyper-parameter settings related to the *NavACL-Q* algorithm.

| *NavACL-Q* Hyperparameters | |
|---|---|
| **Parameter** | **Value** |
| Batch size $m$ | 16 |
| Upper-confidence coefficient for *easy* task $\beta$ | 1.0 |
| Upper-confidence coefficient for *frontier* task $\gamma$ | 0.1 |
| Additional threshold for *easy* task $\chi$ | 0.95 |
| Maximal number of trials to generate a task $n_T$ | 100 |
| Learning rate for $f_\pi$ | $4 \times 10^{-4}$ |

**Appendix C. Arena Randomization**

In this part, we show the randomization for each training arena cells including the initial pose of mobile robot and the target dolly in Table A3. For instance, the initial Yaw-Rotation (either robot or dolly) of 0° corresponds to alignment with the y-axis illustrated by Figure 6 (frontal robot camera heads towards the dolly), and −90° corresponds to alignment with the x-axis (frontal camera points towards the right wall).

**Table A3.** Summary of the task randomization, including the initial pose of AVG, the pose of the target dolly and obstacles.

| **Description** | **Randomization** | **Induced Randomization with Respect to Geometric Property** |
|---|---|---|
| Initial Robot Yaw-Rotation | Uniform sampled from the interval $[-90°, 90°]$ | Relative Rotation: [1.5 m, 5 m] |
| Initial Dolly Yaw-Rotation | Uniform sampled from the interval $[-15°, 15°]$ | |
| Number of Obstacles | 1 to 4 | Agent Clearance/ Goal Clearance: [2 m, 8 m] |
| Position of Obstacles | Randomly placed left and right of the dolly, with a distance uniformly sampled from the interval $[2\,\text{m}, 5\,\text{m}]$ | |
| Initial Robot Position | −0.5 m to 0.5 m on y- and x-axis | Agent-Goal Distance: [1.5 m, 5 m] |
| Initial Dolly Position | Uniformly sampled from a circle segment with radius = 5 m and central angle 30°, where the center of the segment corresponds to the center of the robot, with minimum 1.5 m distance to the robot | |

**Appendix D. Mobile Robot and Target Dolly Specification**

In Table A4, we show the specification of mobile robots and the target dolly. One can see that the goal state for the mobile robot is strict and therefore posing great challenges to DRL algorithms.

**Table A4.** Technical details of the mobile robot and the target dolly.

| Mobile Robot | |
|---|---|
| Length, Width, Height | 1273 mm × 630 mm × 300 mm |
| Maximum Speed | 1.2 m/s |
| LiDAR Sensor | 2× 128 Beams, each FOV 225°, Max Distance: 6 m |
| Frontal RGB Camera | 80 × 80 × 3 pixel , FOV 47° |
| **Dolly** | |
| Length, Width | 1230 mm × 820 mm |

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
