# Peer review of "Using Deep Reinforcement Learning with Automatic Curriculum Learning for Mapless Navigation in Intralogistics"

_applsci, doi:10.3390/app12063153_

Round 1

Reviewer 1 Report

  1. The authors were recommended to provide full form for each abbreviation when they first appeared in the manuscript, such as RGB, NVIDIA, etc.
  2. Please provide additional explanations for the multi-modal sensor readings in the statement.
  3. Each equation in the manuscript should be clearly cited in the study.
  4. Please illustrate Eq. (2) with more explanations.
  5. The following studies were recommended to be properly cited in the study: [1] Probabilistic Indoor Positioning and Navigation (PIPN) of Autonomous Ground Vehicle (AGV) Based on Wireless Measurements, IEEE Access, vol. 9, pp. 25200-25207, 2021. [2]Traffic flow prediction by an ensemble framework with data denoising and deep learning model, Physica A: Statistical Mechanics and its Applications, vol. 565, p. 125574, 2021.

Author Response

Comments and Suggestions for Authors

  • The authors were recommended to provide full form for each abbreviation when they first appeared in the manuscript, such as RGB, NVIDIA, etc

We appreciate the reviewer’s kind recommendation to add full form for each abbreviation. In our current version, we have appended an abbreviation list in page 24 of the manuscript according to the template. For modifications, we also updated the abbreviation list with your suggested terms in line 60. However, ‘NVIDIA’ is already the full name. 

  • Please provide additional explanations for the multi-modal sensor readings in the statement.

We thank the reviewer for this point. In our case, multi-modal sensor readings mean that the agent both received the frontal RGB visual readings as well as two Lidar readings with each providing 128 beams and covering the range of 225 degree. The detailed setting is present in Table A4. We add a brief description first to from line 50~54 in Introduction and also more detailed explained in Caption of Figure 1.

  • Each equation in the manuscript should be clearly cited in the study.

We thank the reviewer for potential improvements on the paper and we have made the corresponding adjustments for all equations in our manuscript.

  • Please illustrate Eq. (2) with more explanations.

We have provided more explanations in line 308~317. Thanks for the point!

  • The following studies were recommended to be properly cited in the study: [1] Probabilistic Indoor Positioning and Navigation (PIPN) of Autonomous Ground Vehicle (AGV) Based on Wireless Measurements, IEEE Access, vol. 9, pp. 25200-25207, 2021. [2]Traffic flow prediction by an ensemble framework with data denoising and deep learning model, Physica A: Statistical Mechanics and its Applications, vol. 565, p. 125574, 2021.

We appreciate the reviewers’ recommendations for related work and have added both of the related works in line 147~151 and line 170.

Reviewer 2 Report

Please refer to the attachment for detailed comments

Author Response

We greatly appreciate reviewer’s positive impression on our work and our answers to your questions are listed below.

Point 1: Firstly, I'm confused about the scene part, including whether the number of training scenarios and verification scenarios is sufficient, why to choose the warehouse scene, and whether there are some features in the scene that affect the output, especially objects or special shapes.

  • Why to choose warehouse scene? 

The entire goal of the project with KION Group Ag. is to let autonomous guided vehicle to perform autonomous dockering tasks in warehouse scenarios. This work is a part of the entire project, and aims to investigate deep reinforcement learning approaches for intralogistics navigation in a warehouse setting. However, our approach is not only restricted to a warehouse setting, but can also be extended to other goal-reaching navigation tasks in other domains.

  • Whether the number of training scenarios and verification scenarios is sufficient? 

In our current settings, we parallelize 9 threads for experience collection first for speedup and also generalization of the learned policy. The 9 agents collect the experience in 9 different cells as shown in Figure 3, and for each agent/thread, the initial pose of mobile robot and goal dolly pose are also randomized for each episode to avoid overfitting the policy. Each cell is also different from the rest in light source placement, wall and ground colors, patterns, obstacle positions and etc. to increase generalizability. We agree that increasing the number of different training scenarios can further boost the generalizability of the learned policy. However, increasing the number of agents and training cells in a map will slow down the simulation and result in an overall long training cycle, as Nvidia Isaac Simulation is also realistic. We also checked the number of training scenarios in our experimental processes and found 9 training scenarios was optimal for the performance of our machine. For validation scenarios, please refer to our answer in Point 2.

  • Whether there are some features in the scene that affect the output, especially objects or special shapes? 

This is an interesting question, and the answer is yes, there are definitely some features affecting vehicle’s decision. For instance, the appearance of the target dolly into the frontal RGB camera will definitely help guide the autonomous vehicle towards the goal. So is the case for collision avoidance of the obstacles. This can be seen from our supplementary videos. However, it should be noted that the agent’s decision is jointly affected by RGB camera and LiDAR readings, not solely from RGB camera. We found that if the agent navigates from farther distances, which exceeds the distance used in training. Even if the agent can capture goal in RGM camera, it behaves very sub-optimally, making circles but gradually approaching the target. But as long as the agent enters the region sufficiently close to the dolly, i.e., within the scope of distance in training, it behaves near-optimally. This is a sign showing that the LiDAR reading also affect the decisions. We have mentioned this phenomenon in the Section Discussion.

Point 2:  Secondly, maybe there is a lack of comparisons in the results tested on the validation set, especially intuitive trajectory plots on the scenery and error analysis, etc. Intuitive results may help readers better understand the strengths of this paper.

We greatly appreciate the reviewer for the suggestions of adding more intuitive results as improvements. In the modified version, we integrated some detailed statistics in Section Results, especially adding a new table (Table 3) showing measurable differences on our 3 DRL variants and baseline approach provided by Nvidia, shown in page 18. We also further expand the result presentations respectively in line 591~599, 650~652, 657~660 and 684~697. For more intuitive understanding on our results, we place these intuitive results also in Abstract (line 11~15) and Conclusion (line 840~ 848).

Regarding the comparison in the results tested on the validation set, we believe that a systematic evaluation on our validation set for each ablation variant has been conducted.  As shown in Section 4.2, we create a validation set which not seen in the training scenarios, and would like to check the performance of the trained policy from different ablation variants (NavACL-Q p.t. ,  NavACL-Q e.t.e and RND). To do so, we define a grid test for different initial position of the vehicle, described as (x, y, orientation), which corresponds to 11(x)*11(y)*8(orientation) = 968 configurations. The initial state in validation already extrapolates the ones used in training, i.e., larger absolute orientation angles. For each configuration, i.e., each dot in Figure 7, we performed 9 runs, resulting in the total 11*11*8*9 = 8712 runs for each ablation variant. In this case, we believe the comparisons on the validation results should be sufficient. The trajectory plots are just intended to present the readers with an intuitive result on the trajectory learned, as the comprehensive quantitative results are illustrated in Section 4. We apologize for the potential confusion brought to the reviewer and have added one explicit sentence in line 531.

Point 3: Thirdly, although the language does not seem wrong, it may be less readable for readers. Authors should try to clarify the logic of each section, clearly explain the method used in this paper, and explain the significance of doing so. Besides, a large number of abbreviations appear in the article, and for many readers who do not know enough about the field, these abbreviations are complicated to read. Please double-check that all abbreviations are necessary to use and explain them as clearly as possible. I have noticed that the author has made an abbreviated index. It is good, and please give a hint to the reader in the text.

We thank the reviewer for the potential suggestions on the improvement of our script. We have improved our explanations for a better reading in each Section, especially for the result part line 456~ 469, and discussion part line 701~ 704, making the structure clearer. We also expand the explanation for Equation 2 in Methods Section, line 308~317. The number of abbreviations is also reduced accordingly. We currently make a hint of an abbreviated index in Section 4.1, line 481.

Reviewer 3 Report

The article is well-written and covers and interesting topic. In fact, my recommendation is that the article is accepted in present format.

Each section is well written and have a nice flow - I do not feel any changes are needed to improve them. 

One minor comment - it would be beneficial to include quantitative values in the abstract, on lines 9-11 where the authors state that their agent outperforms, so readers can understand the measurable difference. This is also the same comments for line 13 regarding the training speed. However, this can be done at the authors' discretion and the article is already publishable.

Author Response

  • One minor comment - it would be beneficial to include quantitative values in the abstract, on lines 9-11 where the authors state that their agent outperforms, so readers can understand the measurable difference. This is also the same comments for line 13 regarding the training speed. However, this can be done at the authors' discretion and the article is already publishable.

We appreciate the reviewer’s great review and positive feedbacks! We have added some direct statistics in abstract line 11~15 to let others understand a measurable difference. 

Round 2

Reviewer 1 Report

I think we can accept the manuscript now. 

Reviewer 2 Report

The authors answer all the questions I asked.

Please consider adding the following related references.
Hammad S. Alhasan, Patrick C. Wheeler, Daniel T. P. Fong, "Application of Interactive Video Games as Rehabilitation Tools to Improve Postural Control and Risk of Falls in Prefrail Older Adults", Cyborg and Bionic Systems, vol. 2021, Article ID 9841342, 11 pages, 2021. 
Classifying Motion Intention of Step Length and Synchronous Walking Speed by Functional Near-Infrared Spectroscopy